# Relative humidity gradients as a key constraint on terrestrial water and energy fluxes

Yeonuk Kim[1], Monica Garcia[2], Laura Morillas[3], Ulrich Weber[4], T. Andrew Black[5], Mark S. Johnson[1,3,6]

[1]Institute for Resources, Environment and Sustainability, University of British Columbia, Vancouver, V6T1Z4, Canada.
[2]Department of Environmental Engineering, Technical University of Denmark, Lyngby, 2800, Denmark.
[3]Center for Sustainable Food Systems, University of British Columbia, Vancouver, V6T1Z4, Canada.
[4]Max Planck Institute for Biogeochemistry, Hans Knoell Strasse 10, 07745 Jena, Germany
[5]Faculty of Land and Food Systems, University of British Columbia, Vancouver, V6T1Z4, Canada.
[6]Department of Earth, Ocean and Atmospheric Sciences, University of British Columbia, Vancouver, V6T1Z4, Canada.

*Correspondence to*: Yeonuk Kim (yeonuk.kim.may@gmail.com)

**Abstract.** Earth's climate and water cycle are highly dependent on terrestrial evapotranspiration and the associated flux of latent heat. Although it has been hypothesized for over 50 years that land dryness becomes embedded in atmospheric conditions through evaporation, underlying physical mechanisms for this land-atmospheric coupling remain elusive. Here, we use a novel physically-based evaporation model to demonstrate that near-surface atmospheric relative humidity (*rh*) fundamentally coevolves with *rh* at the land surface. The new model expresses the latent heat flux as a combination of thermodynamic processes in the atmospheric surface layer. Our approach is similar to the Penman-Monteith equation but uses only routinely measured abiotic variables, avoiding the need to parameterize surface resistance. We applied our new model to 212 in-situ eddy covariance sites around the globe and to the FLUXCOM global-scale evaporation product to partition observed evaporation into diabatic vs. adiabatic thermodynamic processes. Vertical *rh* gradients were widely observed to be near zero on daily to yearly time scales for local as well as global scales, implying an emergent land-atmosphere equilibrium. This equilibrium allows for accurate evaporation estimates using only the atmospheric state and radiative energy, regardless of land surface conditions and vegetation controls. Our results also demonstrate that the latent heat portion of available energy (i.e., evaporative fraction) at local scales is mainly controlled by the vertical *rh* gradient. By demonstrating how land surface conditions become encoded in the atmospheric state, this study will improve our fundamental understanding of Earth's climate and the terrestrial water cycle.

## 1 Introduction

Latent heat flux (*LE*) associated with plant transpiration and evaporation from soil and intercepted water (i.e., evapotranspiration, ET) links the water cycle with the terrestrial energy budget. More than half of the incoming radiation energy at the land surface is consumed as *LE*, making ET the second largest flux in the terrestrial water balance after

precipitation (Oki and Kanae, 2006). Also, *LE* is a controlling factor for near-surface climatic conditions such as temperature and humidity (Ma et al., 2018;Byrne and O'Gorman, 2016). While most research has been devoted to developing and improving rate-limiting parameters constraining *LE* (e.g., García et al., 2013;Martens et al., 2017), exploring the governing physics of *LE* has received less attention following earlier pioneering work (Schmidt, 1915;Penman, 1948;Bouchet, 1963;Monteith, 1965;Priestley and Taylor, 1972). Nevertheless, improvement of the theoretical understanding of *LE* still remains an essential cornerstone to correctly simulate and predict climate and hydrological cycles (Emanuel, 2020).

Climatic conditions over the land surface are getting not only warmer but also drier in recent decades (i.e., decrease in relative humidity) (Sherwood and Fu, 2014;Willett et al., 2014;Byrne and O'Gorman, 2018), but land-atmosphere feedback processes shaping the near-surface atmospheric state are still not well understood. In the early 1960's, Bouchet (1963) hypothesized that land surface dryness is coupled to the atmospheric state through *LE*, with the Bouchet hypothesis now widely accepted (Ramírez et al., 2005;Fisher et al., 2008;Mallick et al., 2014). However, the underlying physical mechanisms for this land-atmospheric coupling still remain elusive (McNaughton and Spriggs, 1989). Recently, McColl et al. (2019) introduced a novel theoretical perspective on land-atmosphere coupling which is referred to as "Surface Flux Equilibrium (SFE)". They hypothesized that relative humidity (*rh*) reaches a steady state value in an idealized atmospheric boundary layer at daily to monthly timescale. Under steady *rh* conditions (i.e., the SFE state), *LE* can be determined using only the atmospheric state and radiative energy. Although this method performed well compared to actual *LE* observations for inland continental sites (McColl and Rigden, 2020;Chen et al., 2021), a further investigation is needed to understand how dynamics of turbulent heat fluxes in the atmospheric surface layer at sub-daily time scale evolve to the SFE state.

A traditional way to express the atmospheric surface layer processes is to partition *LE* into diabatic and adiabatic processes using the Penman-Monteith (PM) equation (Monteith, 1965), as proposed by Monteith (1981). The PM equation combines the energy balance equation with mass-transfer theory for water vapour and sensible heat, resulting in diabatic (radiative energy-related) and adiabatic (vapour pressure deficit-related) processes for a parcel of air in contact with a saturated surface (Monteith, 1981).

$$LE = \underbrace{\frac{S}{S+\gamma(\frac{r_a+r_s}{r_a})} \cdot Q}_{Diabatic\ process} + \underbrace{\frac{\rho c_p}{S+\gamma(\frac{r_a+r_s}{r_a})} \cdot \frac{e^*(T_a)-e_a}{r_a}}_{Adiabatic\ process} \qquad (1)$$

where $S$ is the linearized slope of saturation vapour pressure versus temperature (hPa K$^{-1}$), $\gamma$ is the psychrometric constant (hPa K$^{-1}$), $\rho$ is the air density (kg m$^{-3}$), $c_p$ is the specific heat capacity of air at constant pressure (MJ kg$^{-1}$ K$^{-1}$), and $Q$ is available radiative energy (i.e., the difference between net radiation ($R_n$) and soil heat flux ($G$) and expressed in units of W m$^{-2}$). $e^*(T_a)$ is the saturation vapour pressure (hPa) corresponding to the air temperature ($T_a$) measured at a reference height (typically 2 m or eddy flux measurement height), and $e_a$ is vapour pressure (hPa) at the reference height. $e^*(T_a)$- $e_a$ is known as atmospheric vapour pressure deficit (VPD, expressed in units of hPa). $r_a$ is aerodynamic resistance to heat and water vapour transfer (s m$^{-1}$), and $r_s$ is surface resistance to water vapour transfer (s m$^{-1}$) representing drying soil and/or plant stomatal closure.

In principle, high $Q$ and VPD at the reference height increase the diabatic and the adiabatic terms respectively in the PM equation, and as such, $Q$ and VPD are the two primary drivers of evaporation (Monteith and Unsworth, 2013). Yet, this "high VPD leads to high $LE$" interpretation cannot be generalized because $r_s$ increases with VPD due to stomatal closure by vegetation under high VPD conditions (Tan et al., 1978;Novick et al., 2016;Massmann et al., 2019). While the PM equation is useful to explore biological control of $LE$ through $r_s$ (Jarvis and McNaughton, 1986;Peng et al., 2019), physical mechanisms corresponding to each term in Eq. (1) are less intuitive due to the sensitivity of $r_s$ to VPD. As a result, how the atmospheric state affects evaporation and vice versa remains ambiguous in the PM equation.

Is there a way to mathematically express the physical mechanisms of $LE$ without requiring $r_s$? In this paper, we present a pair of equations expressing actual $LE$ as a combination of diabatic and adiabatic processes without requiring $r_s$. Similar to the PM equation, our new equations are derived by combining the energy balance equation with the flux gradient equations but crucially ours do not include $r_s$. The novel equations are applied empirically to eddy-covariance observation sites and a global $LE$ dataset to explore land-atmosphere coupling processes at various spatiotemporal scales. To do this, we decomposed observed $LE$ into adiabatic and diabatic components and discuss how these patterns can help to understand land-atmosphere interactions and potential responses under future climatic conditions.

## 2. Theory

### 2.1. A pair of evaporation equations for an unsaturated surface

In this section, we derive a pair of evaporation equations for an unsaturated surface. In this derivation, we assume a horizontally homogenous landscape where the sources of water vapour and heat are identical. Under this idealized condition, aerodynamic resistances ($r_a$) to heat and water vapour transfers are identical. Here, $r_a$ is a parameterization of turbulent mixing due to mechanical turbulence and buoyancy driven by surface heating.

We first express $LE$ using a flux gradient equation as $LE = \frac{\rho c_p}{\gamma} \frac{e_s - e_a}{r_a}$, where $e_s$ is the surface vapour pressure. Here, the subscript $s$ indicates the land surface which is defined as an idealized plane specified as the sum of displacement height and roughness length for heat (Knauer et al., 2018a;Novick and Katul, 2020). If the land surface is saturated, $e_s$ becomes equivalent to the saturation vapour pressure (i.e., $e_s = e^*(T_s)$). For an unsaturated land surface, however, relative humidity should be introduced as $e_s = rh_s e^*(T_s)$, where $rh_s$ is surface relative humidity, i.e., the ratio of $e_s$ to $e^*(T_s)$. For a vegetated surface, $rh_s$ as defined in this study represents relative humidity of the foliage surface and is conceptually equivalent to surface water availability in Li and Wang (2019). For a bare soil land surface, $rh_s$ represents soil surface relative humidity which can be found using the "alpha" method that is parameterized using soil moisture content or soil water potential (Lee and Pielke, 1992;Wu et al., 2000;Cuxart and Boone, 2020). Using $rh_s$, $LE$ can be written as $LE = \frac{\rho c_p}{\gamma} \frac{rh_s e^*(T_s) - e_a}{r_a}$ for an unsaturated surface condition.

In order to decompose *LE* into two individual fluxes related to temperature and relative humidity gradients, we add $-rh_s e^*(T_a) + rh_s e^*(T_a)$ (or $-rh_a e^*(T_s) + rh_a e^*(T_s)$) to the numerator of the flux gradient equation and rewrite *LE* as follows:

$$LE = \frac{\rho c_p}{\gamma} rh_s \frac{e^*(T_s) - e^*(T_a)}{r_a} + \frac{\rho c_p}{\gamma} e^*(T_a) \frac{rh_s - rh_a}{r_a} \tag{2}$$

$$LE = \frac{\rho c_p}{\gamma} rh_a \frac{e^*(T_s) - e^*(T_a)}{r_a} + \frac{\rho c_p}{\gamma} e^*(T_s) \frac{rh_s - rh_a}{r_a} \tag{3}$$

We then approximate $e^*(T_s) - e^*(T_a) = S(T_s - T_a)$ using the saturation vapour pressure slope at the air temperature (*S*), and introduce a flux gradient equation for sensible heat flux (i.e., $H = \rho c_p \frac{T_s - T_a}{r_a}$) into Eqs. (2) and (3). Then, the energy balance equation is combined to substitute *H* with $Q - LE$. As results, *LE* can be expressed as follows:

$$LE = \underbrace{\frac{rh_s S}{rh_s S + \gamma} \cdot Q}_{Diabatic\ process:\ LE_Q} + \underbrace{\frac{\rho c_p e^*(T_a)}{rh_s S + \gamma} \cdot \frac{rh_s - rh_a}{r_a}}_{Adiabatic\ process:\ LE_G} = LE_Q + LE_G \tag{4}$$

$$LE = \underbrace{\frac{rh_a S}{rh_a S + \gamma} \cdot Q}_{Diabatic\ process:\ LE_{Q'}} + \underbrace{\frac{\rho c_p e^*(T_s)}{rh_a S + \gamma} \cdot \frac{rh_s - rh_a}{r_a}}_{Adiabatic\ process:\ LE_{G'}} = LE_Q' + LE_G' \tag{5}$$

where $LE_Q$ (and $LE_Q'$) is a diabatic component, $LE_G$ (and $LE_G'$) is an adiabatic component of latent heat flux. While the diabatic component is mainly determined by available energy (*Q*), the adiabatic component is driven by turbulent mixing and vertical gradient of *rh*. Monteith (1981) originally suggested an equation equivalent to Eq. (4) for the case when the surface does not reach saturation. To our knowledge, Eq. (5) is derived for the first time here. Equations (4) and (5) include $rh_s$ to compensate for eliminating $r_s$ from the original PM equation.

Since the adiabatic process in Eqs. (4) and (5) are controlled by the vertical difference of *rh*, we refer to Eqs. (4) and (5) as the proposed PM$_{rh}$ model (Penman-Monteith equation expressed using *rh*) to distinguish it from the original PM model. The two equations (4) and (5) are complementary to each other in that they represent distinct thermodynamic paths, each of which will be discussed in the next section. Arguably, applying PM$_{rh}$ can provide new insights into the fundamental mechanisms of *LE*, particularly when it is decomposed into its diabatic component ($LE_Q$ or $LE_Q'$) and its adiabatic component ($LE_G$ or $LE_G'$). In the following sections, we will discuss theoretical meanings of Eqs. (4) and (5) in-depth.

## 2.2. Generalized Penman equation

Before discussing PM$_{rh}$ in-depth, we revisit the Penman equation (Penman, 1948) to help with the physical reasoning behind our proposed framework. The widely recognized form of the Penman equation, which was developed as an *LE* model for a saturated surface, is as follows:

$$LE = \underbrace{\frac{S}{S + \gamma} \cdot Q}_{Diabatic\ process} + \underbrace{\frac{\rho c_p [e^*(T_a) - e_a]}{[S + \gamma] r_a}}_{Adiabatic\ process} \tag{6}$$

We rearrange this formulation to derive Eq. (7) by factoring out $e^*(T_a)$ and introducing $rh_a = \frac{e_a}{e^*(T_a)}$ into the second term.

$$LE = \underbrace{\frac{S}{S+\gamma} \cdot Q}_{Diabatic\ process} + \underbrace{\frac{\rho c_p e^*(T_a)}{S+\gamma} \cdot \frac{1-rh_a}{r_a}}_{Adiabatic\ process} \tag{7}$$

Equations (6) and (7) are mathematically equivalent, but their interpretations are quite different. In Eq. (6), the adiabatic process is controlled by VPD at the reference height. However, in Eq. (7), the adiabatic process acts over the vertical $rh$ gradient, i.e. the difference in $rh$ from the surface to the reference height ($rh_a$). Since the Penman equation is a model for saturated surfaces, $1 - rh_a$ in Eq. (7) indicates the difference in $rh$ over the vertical distance between the ground surface and the reference height. Arguably, Eq. (7) is more thermodynamically sound compared to Eq. (6) since $rh$ is an ideal-gas approximation to the water activity (Lovell-Smith et al., 2015) which represents the chemical potential of water ($\mu_w$) (Monteith and Unsworth, 2013;Kleidon and Schymanski, 2008). When the vertical gradient of $rh$ dissipates owing to well-developed turbulence, the land surface and the atmosphere are in thermodynamic equilibrium (Kleidon et al., 2009). Therefore, taking Eq. (7) instead of Eq. (6) allows us to view the adiabatic process of the Penman model as an equilibration process driving land-atmosphere equilibrium by bringing the surface $\mu_w$ to that of the atmosphere.

As with our interpretation of the Penman model, we can view Eqs. (4) and (5), as a generalized form of the Penman model. Here, the $LE_G$ (or $LE_G{}'$) term is an equilibration process between the land and the atmosphere when the land surface is not saturated. It is worth noting that $LE_G$ can be negative when $rh_s$ is less than $rh_a$. Thus, the $LE_G$ term operated by turbulent mixing acts to reduce the vertical $rh$ gradient. This physical interpretation is consistent with recent findings that the variance of the $rh$ gradient tends to be minimized over the course of the day, implying that the difference between $rh_s$ and $rh_a$ is reduced (Salvucci and Gentine, 2013;Rigden and Salvucci, 2015). The diabatic $LE_Q$ (or $LE_Q{}'$) term can be understood as equilibrium $LE$ for an unsaturated surface, which we discuss later in section 2.4.

### 2.3. Thermodynamic paths

How can we interpret the two formulas of PM$_{rh}$ in Eqs. (4) and (5)? To explain the two forms, the psychrometric relationship is applied to a parcel of air near an unsaturated land surface that is under constant pressure and steadily receiving radiation energy. The psychrometric diagram in Fig. 1 describes the magnitude of turbulent flux (where the length of the arrow corresponds to the magnitude) from the view point of a parcel of air located at a reference height (an approach based on work by Monteith (1981)). Since the parcel of air receives heat and water vapour from the land surface, the final state is represented by the surface condition, while the initial state is represented by the atmospheric conditions at the reference height. Therefore, the initial thermodynamic state of the air parcel can be represented by its temperature and water vapour pressure such as point A in Fig. 1. The initial state is changed by two processes as follows: (1) equilibrating between the land surface ($rh_s$) and the air parcel ($rh_a$), and (2) increasing enthalpy forced by the incoming energy. It should be noted that the changing process (i.e., thermodynamic path) from the initial to the final states in this discussion should be understood as the magnitude of the turbulent heat fluxes.

In the $rh$ equilibrating process, the air parcel is adiabatically cooled (or heated when $rh_s < rh_a$) due to turbulent mixing, while the enthalpy of the parcel is not changed. Therefore, the increase (decrease) in latent heat content in the parcel is exactly

balanced by a decrease (increase) in sensible heat ($A \rightarrow B$ in Fig. 1: trajectory along constant enthalpy line). This process is equivalent to the $LE_G$ term in Eq. (4). Now, the air parcel is in thermodynamic equilibrium with the land surface (point B in Fig. 1). Then, the air parcel receives energy while the equilibrium is sustained (i.e., $rh_s$ is steady), which increases both the temperature and absolute water vapour content of the air parcel ($B \rightarrow C$ in Fig. 1). This process can be expressed as $LE_Q$ of Eq. (4). Consequently, the thermodynamic state of the air parcel approaches point C in Fig. 1.

However, we should recognize that temperature and vapour pressure are "state" variables meaning that they do not depend on the thermodynamic path by which the system arrived at its final state (Iribarne and Godson, 2012). In the above example, we conceptually followed the adiabatic process first and then the diabatic process (Path 1 in Fig. 1), but one can imagine the opposite order. If we choose Path 2 in Fig. 1, the diabatic process comes first, and thus $rh_a$ instead of $rh_s$ is preserved while enthalpy increases (i.e., $LE_Q'$), and the adiabatic process is followed at temperature of $T_S$ (i.e., $LE_G'$). Path 2 is described by Eq. (5).

Therefore, one can interpret the two forms of $PM_{rh}$ in Eqs. (4) and (5) as two thermodynamic paths where the diabatic and adiabatic processes occur simultaneously. It should be noted that the diabatic and adiabatic processes in $PM_{rh}$ are "path" functions and thus they vary by path. For instance, $LE_Q$ is slightly higher than $LE_Q'$ when $rh_s > rh_a$. Also, the absolute magnitude of $LE_G'$ is always bigger than that of $LE_G$ when $Q > 0$ (i.e., vector $B' \rightarrow C$ is longer than vector $A \rightarrow B$ in Fig. (1)).

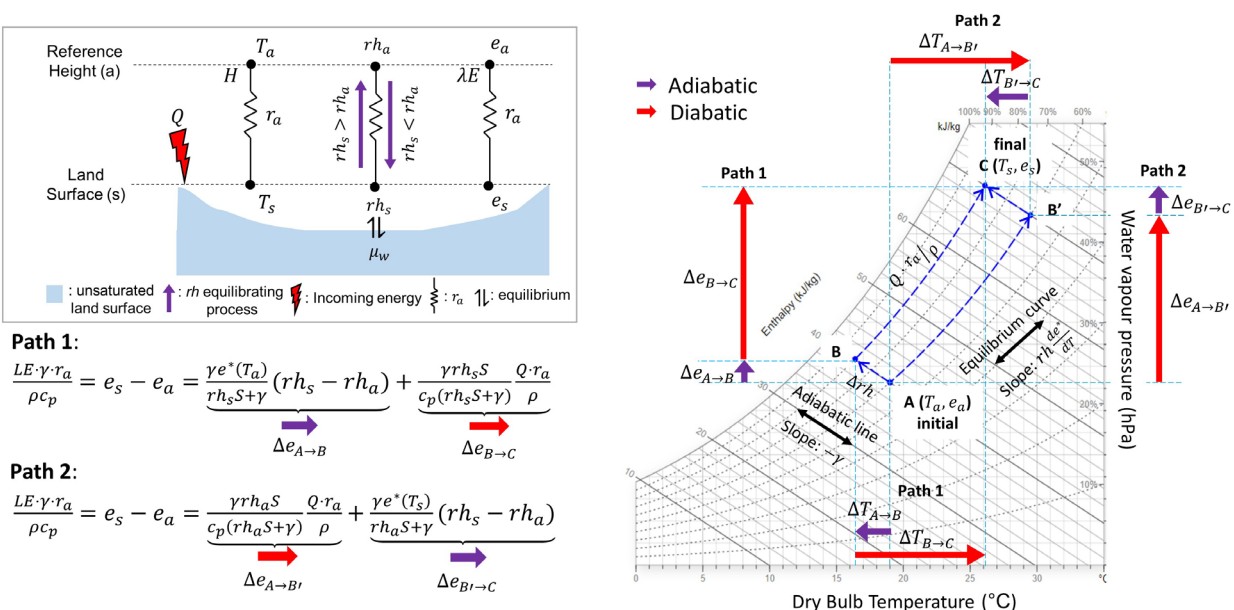

**Path 1:**
$$\frac{LE \cdot \gamma \cdot r_a}{\rho c_p} = e_s - e_a = \underbrace{\frac{\gamma e^*(T_a)}{rh_s S+\gamma}(rh_s - rh_a)}_{\Delta e_{A \rightarrow B}} + \underbrace{\frac{\gamma rh_s S}{c_p(rh_s S+\gamma)}\frac{Q \cdot r_a}{\rho}}_{\Delta e_{B \rightarrow C}}$$

**Path 2:**
$$\frac{LE \cdot \gamma \cdot r_a}{\rho c_p} = e_s - e_a = \underbrace{\frac{\gamma rh_a S}{c_p(rh_a S+\gamma)}\frac{Q \cdot r_a}{\rho}}_{\Delta e_{A \rightarrow B'}} + \underbrace{\frac{\gamma e^*(T_s)}{rh_a S+\gamma}(rh_s - rh_a)}_{\Delta e_{B' \rightarrow C}}$$

Figure 1: Schematic conceptualization of the $PM_{rh}$ model and psychrometric relationship of $PM_{rh}$. The example psychrometric chart is modified from drajmarsh.bitbucket.io/psychro-chart2d.html. Path 1 represents Eq. (4) divided by $\frac{\rho c_p}{\gamma r_a}$ while Path 2 represents Eq. (5) divided by $\frac{\rho c_p}{\gamma r_a}$. Here, the enthalpy change of the air parcel is defined as $\frac{Q \cdot r_a}{\rho}$ (kJ kg$^{-1}$). It should be noted that the difference between the initial and the final states represent the magnitude of the turbulent heat fluxes instead of changes in atmospheric state.

### 2.4. Equilibrium *LE* for an unsaturated surface

Another distinct characteristic of the PM$_{rh}$ model is the way it defines equilibrium at the land-atmosphere interface. Unlike many previous studies which focused on the steady state of VPD (McNaughton and Jarvis, 1983;Priestley and Taylor, 1972;Raupach, 2001), land-atmosphere equilibrium is achieved in the PM$_{rh}$ model when the vertical *rh* gradient (i.e., the $\mu_w$ gradient) dissipates. That is, if $rh_s \approx rh_a$, then it follows that $LE_G$ (or $LE_G$') is zero and thus *LE* becomes

$$LE \approx \frac{rh_a S}{rh_a S + \gamma} Q \tag{8}$$

We note that Eq. (8) is identical to the SFE theory recently introduced by McColl et al. (2019). They hypothesized that in many continental regions, the near surface atmosphere is in state of equilibrium, where *rh* is steady with time in an idealized atmospheric boundary layer at longer than daily time scales. Equation (8) successfully predicted observed *LE* at daily and monthly time scales for inland regions (McColl and Rigden, 2020;Chen et al., 2021), which implies the vertical *rh* gradient tends to evolve toward zero at longer time scales than sub-daily scale. This is logical in that $LE_G$ itself diminishes the vertical *rh* gradient over the course of a day.

From a different standpoint, if an observed *LE* is bigger or smaller than Eq. (8) at a longer time scale such as monthly, it may indicate that the land surface conditions are not completely embedded in the near-surface atmospheric state due to highly wet or dry land conditions. Therefore, $LE_G$ (or $LE_G$') value and sign at monthly time scales could be a useful indicator reflecting land surface dryness relative to the atmosphere.

When both land surface and atmosphere are saturated (i.e., $rh_s \approx rh_a \approx 1$), equation (8) becomes classical equilibrium *LE* (i.e., $LE \approx \frac{S}{S+\gamma} Q$). This is consistent with one of the classical definitions of equilibrium *LE* that defines equilibrium *LE* as evaporation from a saturated surface into saturated air (Schmidt, 1915;Eichinger et al., 1996;Raupach, 2001;McColl, 2020). Therefore, we can regard Eq. (8) as a generalized equilibrium *LE* for an unsaturated surface.

### 3. Materials and Methods

In the following sections, we present a novel physical decomposition of *LE* from PM$_{rh}$ into $LE_Q$ and $LE_G$ components to aid in understanding the governing physics of *LE*. Also, the proportion of net available energy consumed in evapotranspiration, known as the evaporative fraction ($EF = \frac{LE}{Q}$) is decomposed into $\frac{LE_Q}{Q}$ and $\frac{LE_G}{Q}$. We conducted a detailed diagnostic analysis of the PM$_{rh}$ model using the multi-year record of an eddy covariance (EC) flux observation site located in a wet-dry tropical climate. We also applied the PM$_{rh}$ model to the 212 EC sites represented in the FLUXNET2015 dataset (Pastorello et al., 2020), and to the FLUXCOM global *LE* product (Jung et al., 2019). We describe the local and global datasets and analysis methods here before presenting the results.

### 3.1. In-situ EC flux observation

In-situ half-hourly EC observations used in this study were made from 2015 to 2018 on a ratoon sugarcane farm in the province of Guanacaste, Costa Rica (10°25′07.60″N; 85°28′22.22″W). The site has a wet-dry tropical climate with a dry season from December to March and a median monthly air temperature ranging from 27 °C to 30 °C. The study site experienced a significant drought in 2015 as the lowest precipitation rate in Fig. 2 (b) (Hund et al., 2018;Morillas et al., 2019). The site was irrigated occasionally during dry seasons via furrow irrigation events, except for 2016 when there was no irrigation due to crop replanting. Due to the ratooning practice (i.e., sugarcane cutting each year followed by resprouting without replanting, detailed explanation in the Supplement), the sugarcane growing seasons varied by year, which provided an opportunity to explore distinct and varied combinations of land surface vs. atmospheric aridity conditions.

The measured $LE$ and sensible heat flux ($H$) were quality controlled following Morillas et al. (2019) (details in the Supplement). For the study period, the surface energy balance closure (i.e., $\frac{LE+H}{R_n-G}$) of 30 min data was 86 %, which is typical of high-quality eddy-covariance data sets (Wilson et al., 2002). When canopy height was less than 1 m, the surface energy balance was almost closed (97%), whereas the closure was 83 % when canopy height was higher than 1 m. It is expected that unmeasured canopy and soil heat storages in this site are significant because the sugarcane canopy grew up to 3.6 m tall with a dense canopy. For instance, Meyers and Hollinger (2004) showed that storage term comprised 14 % of net radiation for a maize field with a 3-m canopy height, and 8% of net radiation for a soybean field with a 0.9-m canopy height, implying larger heat storage capacities for taller crop canopies. Also, since our study site is located within a homogenous landscape (Figure S1), horizontal and vertical advective flux divergence and the influence of secondary circulations on the energy balance closure may be marginal (Mauder et al., 2020;Leuning et al., 2012). Therefore, considering the homogenous landscape of the study site as well as a possible significant role of unmeasured canopy and soil heat storages, we did not force the energy closure. Consequently, we defined $Q$ as the sum of $LE$ and $H$ instead of $R_n - G$. In doing so, we in effect attribute the cause of the surface energy imbalance to unmeasured heat storage terms following Moon et al. (2020).

In order to decompose $LE$ into $LE_Q$ and $LE_G$, we first estimated half-hourly aerodynamic resistance ($r_a$) by considering aerodynamic resistance to momentum transfer and the additional boundary layer resistance for heat and mass transfer (or excess resistance) (Thom, 1972;Knauer et al., 2018a).

$$r_a = \frac{ln[\frac{z_r-d}{z_{0m}}]-\psi_h}{ku_*} + 6.2u_*^{-0.67} \tag{9}$$

The first term on the right-hand side of Eq. (9) is the aerodynamic component and the second term is the boundary layer component. Here, $u_*$ is friction velocity, $k$ is the von Kármán constant (0.41), $d$ is the zero-plane displacement height (d = 0.7$z_h$), $z_{0m}$ is the roughness length for momentum ($z_{0m} = 0.1z_h$), $\psi_h$ is the integrated form of the stability correction function. $z_h$ is canopy height based on manual measurements taken during regular maintenance visits. $r_a$ was estimated using bigleaf R package (Knauer et al., 2018a).

By rearranging Eq. (2), $rh_s$ can be calculated using

$$rh_s = \frac{\gamma LE r_a / \rho c_p + e_a}{SH r_a / \rho c_p + e^*(T_a)} \tag{10}$$

Negative $H$ and inaccurate $r_a$ modelling sometimes yielded negative $rh_s$ or values greater than one, especially at nighttime. In these cases, $rh_s$ was assigned the value of one following the approach described in the bigleaf R package (Knauer et al., 2018a). We then estimated $LE_Q$ and $LE_G$ from Eq. (4).

In order to explore the time scale of the covariances for $LE \sim LE_Q$ and $LE \sim LE_G$ in the frequency domain, we applied

wavelet coherence analysis using WaveletComp R package (Roesch and Schmidbauer, 2014). The package is designed to apply the continuous wavelet transform using Morlet wavelet, which is a popular approach to analyze hydrological and micrometeorological datasets (Hatala et al., 2012;Johnson et al., 2013). A total time series of half-hourly decomposed $LE$ for the 4-year measurement period was used to estimate localized coherence and phase angle. The wavelet coherence can be interpreted as the local correlation between two variables in the frequency-time domain (where red indicates high correlation).

A 0° phase angle (arrow pointing right) indicates periods of positive correlation while a 180° phase angle (arrow pointing left) indicates periods of negative correlation.

## 3.2. FLUXNET2015

The daily scale FLUXNET2015 dataset, which includes 212 empirical eddy-covariance flux tower sites around globe (Pastorello et al., 2020), was used in this study. The turbulent heat fluxes, net radiation, soil heat flux, air temperature, relative

humidity, wind speed, friction velocity, and barometric pressure were obtained from the dataset. For this analysis, we only included daily data for periods for which the quality control flag indicated more than 80 % half-hourly data were present (i.e., measured data in general, or good quality gap-filled data in cases of partially missing data).

In order to decompose daily $LE$ into $LE_Q$ and $LE_G$, we estimated daily aerodynamic resistance ($r_a$) by Eq. (11) instead of Eq. (9) since canopy and measurement heights are unknown (Thom, 1972;Knauer et al., 2018a).

$$r_a = \frac{u_*^2}{u(z_r)} + 6.2u_*^{-0.67} \tag{11}$$

where, $u(z_r)$ is reference height wind speed. $r_a$ was estimated using the bigleaf R package (Knauer et al., 2018a) and $rh_s$ was calculated from Eq. (8).

$LE_Q$ and $LE_Q'$ were calculated using $rh_a$ and $rh_s$ following Eqs. (4) and (5), and then $LE_G$ and $LE_G'$ were calculated by subtracting $LE_Q$ and $LE_Q'$ from $LE$. To calculate $LE_Q$ and $LE_Q'$, we define $Q$ as $LE + H$, but it should be noted that this

approach can include systematic uncertainty since the sum of $LE$ and $H$ measured by eddy covariance is typically lower than $R_n - G$ (i.e., conditions referred to as the energy balance closure problem (Wilson et al., 2002)). To investigate the effect of a lack of energy balance closure on resulting $LE$ terms, we provide Fig. S2 that was generated by 1) defining $Q$ as $R_n - G$, and 2) correcting $LE$ and $H$ based on the assumption that the Bowen ratio ($B = H/LE$) is correct (Pastorello et al., 2020).

### 3.3. FLUXCOM

The FLUXCOM dataset (Jung et al., 2019) is a global-scale machine learning ensemble product which upscales FLUXNET observations (Baldocchi et al., 2001) using Moderate Resolution Imaging Spectroradiometer (MODIS) satellite data and reanalysis meteorological data. In this study we used the monthly *LE* FLUXCOM dataset (0.5° resolution) modelled using MODIS and ECMWF ERA5 reanalysis data (Hersbach et al., 2020).

We obtained $Q$ and *LE* from the FLUXCOM output, and air temperature and dewpoint temperature were retrieved
from ERA5 monthly averaged data (2 m height). *rh*, *S*, and γ were calculated from ERA5 data, and then $LE_Q'$ was calculated. $LE_G'$ was then estimated by subtracting $LE_Q'$ from *LE*.

## 4. Results

### 4.1. Decomposition analysis of in-situ EC flux observation

Application results of the $PM_{rh}$ model to the observed *LE* at an irrigated sugarcane farm in Costa Rica are depicted in Fig. 2.
The decomposition analysis of observed *LE* shows that while $LE_Q$ is the major component of *LE*, $LE_G$ variability plays a non-negligible role in seasonal and interannual behaviour of *LE*. In terms of absolute magnitude, $LE_Q$ term can closely approximate *LE*, and $LE_G$ only represents 15% of total evaporation (Fig. 2 (c)). Also, positive coherence between *LE* and $LE_Q$ was strong over the entire period of observation, particularly at diurnal to multiday time scales (0.5~32 days), implying variability of *LE* is largely determined by $LE_Q$ variability (i.e., red colored regions in Fig. 2 (e)).

Although absolute magnitude of $LE_G$ was much smaller than that of $LE_Q$, the interannual variability of $LE_G$ was larger than the interannual variability of $LE_Q$ (Fig. 2 (c)). Furthermore, *LE* and $LE_G$ also had a strong positive correlation in longer time scales (32~365 days) (i.e., red colored regions in Fig. 2 (f)). Unexpectedly, a negative correlation between *LE* and $LE_G$ at diurnal time scale was observed in the wavelet analysis only when the land surface was dry and there was little vegetation (i.e., after harvest) or during a year in which there was no dry season irrigation applied. Also, we found that EF variability is mostly
determined by $LE_G$ variability since the diurnal and seasonal signal of $Q$ is removed from *LE* in EF. Interestingly, the annual mean $LE_G$ was the highest in 2015, a drought year in which $rh_a$ and precipitation were generally lower than for the other years, while the annual mean $LE_G$ was close to zero in 2016 when there was no application of dry season irrigation due to crop replanting.

To explore the diurnal behaviour of decomposed *LE*, we selected different surficial and atmospheric conditions when
$LE_G$ was zero, positive, or negative in Fig. 3. In the 2016 dry season, $LE_G$ was close to zero as a daily average value, as a result of negative daytime and positive nighttime $LE_G$ values due to dry air and dry soil conditions (no irrigation) and an undeveloped vegetation canopy (Fig. 3 (a)). Daily $LE_G$ was also close to zero during wet season conditions (e.g., Fig. 3 (b)). In this case, $LE_G$ was near zero during both daytime and nighttime periods due to near saturated atmospheric and land surface conditions.

These two cases show that "dry land-dry air" or "wet land-wet air" conditions can each lead to daily scale land-atmosphere
equilibrium, although the diurnal pattern of $LE_G$ is starkly different for dry land-dry air vs. wet land-wet air conditions.

Meanwhile, when $rh_a$ was low and the canopy was well-developed, $LE_G$ was found to be positive during both daytime
and nighttime periods (Fig. 3 (c)). On the other hand, during post-harvest conditions when vegetative canopy cover was
minimal and air and soil moisture levels were low, daily $LE_G$ was found to be negative as a result of negative daytime and
positive nighttime $LE_G$ (Fig. 3 (d)). Diurnal variation of $rh_s$ was maximized when daily $LE_G$ was negative, implying a large
diurnal fluctuation in surface temperature under the drier land surface conditions. Regarding the overall diurnal pattern, $LE_G$
generally declined during the morning and increased in the afternoon, which is consistent with the well-known diurnal pattern
of EF (Gentine et al., 2011;Gentine et al., 2007) (Fig. 3 (e)).

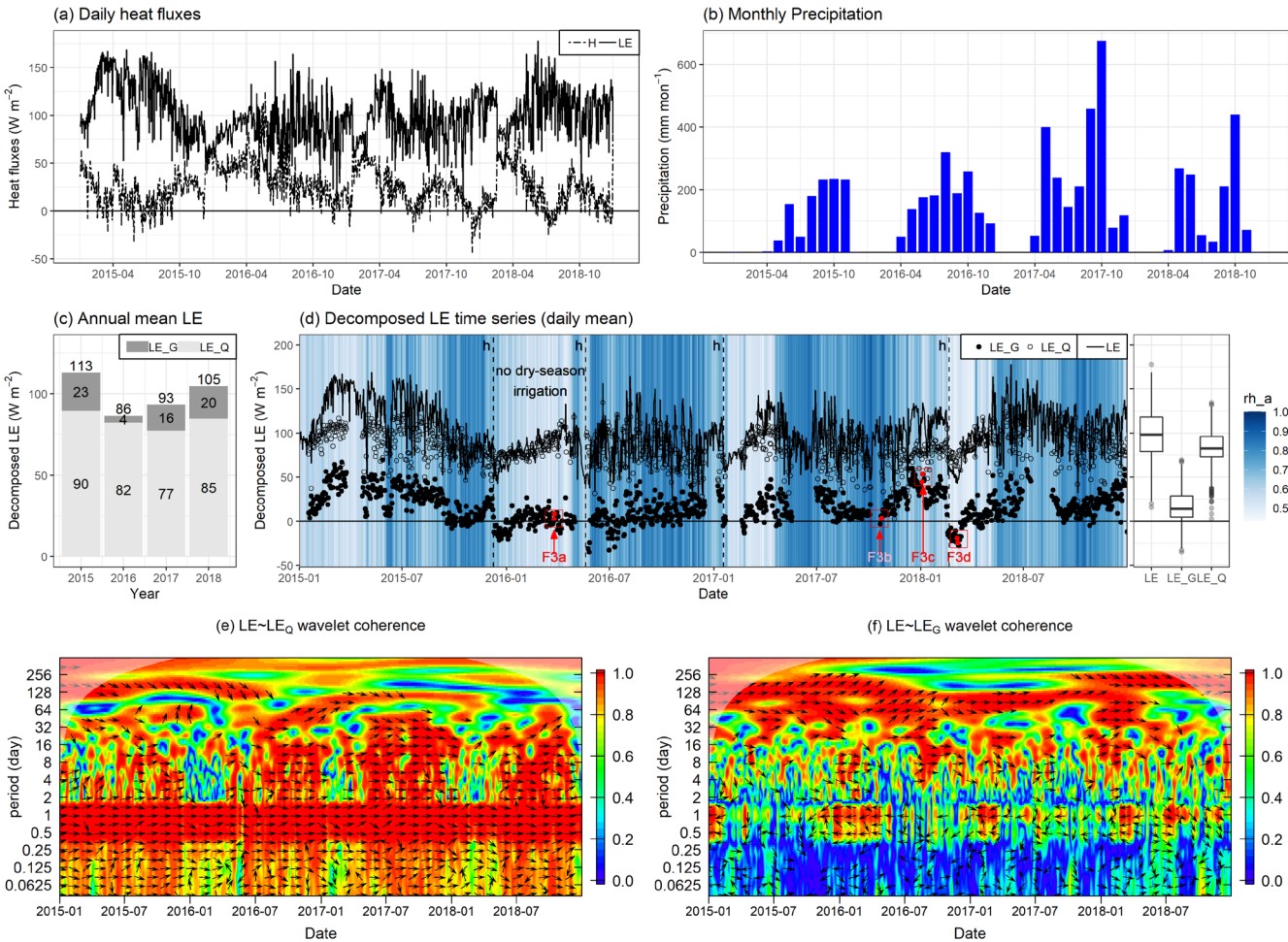

**Figure 2: Time series for the sugarcane EC tower site in Costa Rica. Panel (a) is daily heat fluxes and panel (b) is monthly
precipitation. Panel (c) is mean annual $LE$ and its two components and (d) is time series of $LE$, $LE_Q$, and $LE$ with a background color
of $rh_a$. Dashed lines with "h" in panel (d) indicate sugarcane harvest. Panels (e) and (f) are wavelet coherence of $LE$ with $LE_Q$ and**

*LE* with *LE_G*. Red and blue colors indicate high and low correlation, respectively. Arrows (pointing right: in-phase; left: antiphase) only appear when the coherence is significant (p<0.01).

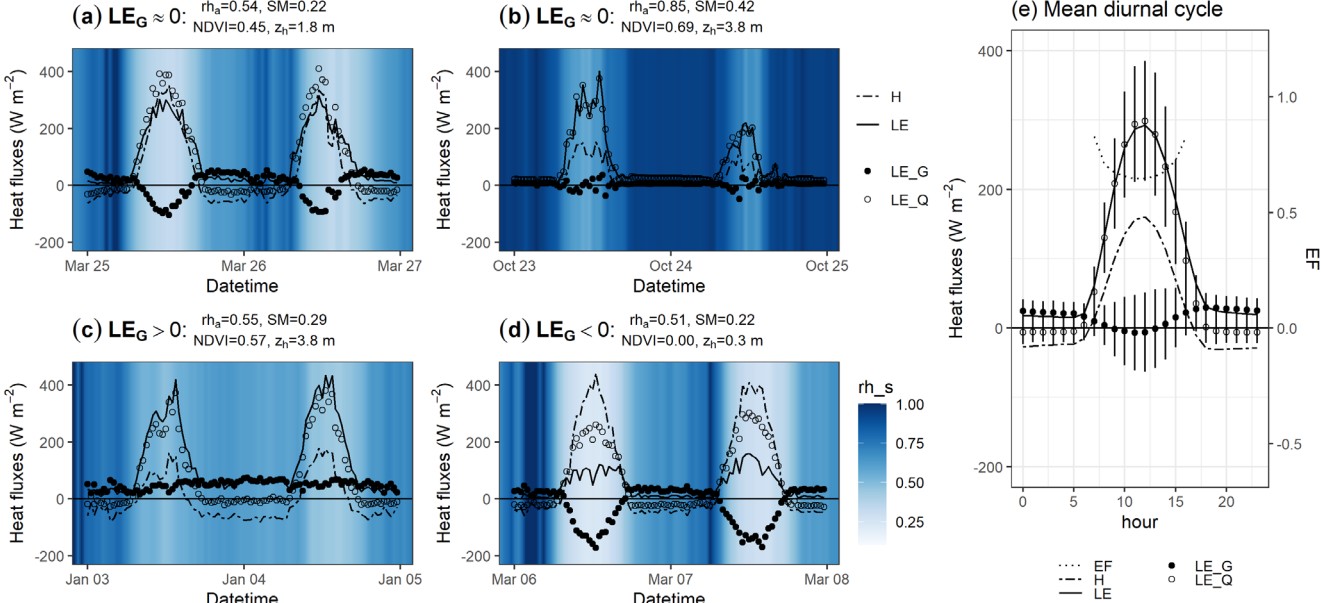

**Figure 3: Half-hourly time series of heat fluxes and the two components indicated in Figure 2 (d). The background color represents *rh_s*. Here, *rh_a* is mean atmospheric relative humidity, SM is volumetric soil water content, NDVI is normalized difference vegetation index, and *z_h* is canopy height. Panel (e) presents the long-term mean diurnal cycle of decomposed *LE* (dots) and EF (dashed line).**


## 4.2. Decomposition analysis of FLUXNET2015 dataset

Decomposition analysis of the daily FLUXNET2015 dataset is illustrated in Fig. 4. In terms of absolute magnitude of each term, the majority of $LE_G$ (and $LE_G'$) values ranged from -50 W m$^{-2}$ to 50 W m$^{-2}$, with some exceptional values approaching ±100 W m$^{-2}$. On the other hand, $LE_Q$ (and $LE_Q'$) values ranged from 0 W m$^{-2}$ to 150 W m$^{-2}$.

One of the interesting findings from the decomposition analysis of FLUXNET2015 dataset was that differences between $LE_Q$ and $LE_Q'$ are marginal at a daily time scale (the slope is close to one in Fig. 4 (a1)). This result implies that although the diabatic processes expressed by Eqs. (4) and (5) are different in magnitude due to the difference between $rh_s$ and $rh_a$ (see section 2.3), these differences are practically negligible. This is an important point since $LE_Q'$ can be determined simply and directly using by reference height meteorological measurements, while $LE_Q$ is required to know $rh_s$.

As for the adiabatic terms, $LE_G'$ is roughly 1.1 times $LE_G$ at a daily time scale (Fig. 4 (a2)), which is consistent with the theory regarding their respective thermodynamic paths. As we discussed in section 2.3, the absolute magnitude of $LE_G'$ must be bigger than that of $LE_G$ when available energy is positive. Therefore, the empirical relationship between $LE_G$ and $LE_G'$ in Fig. 4 (a2) is a consequence of a physical principle, and this result may provide the following empirical relationship:

$$1.1 \frac{\rho c_p e^*(T_a)}{rh_s S + \gamma} \cdot \frac{rh_s - rh_a}{r_a} \approx \frac{\rho c_p e^*(T_s)}{rh_a S + \gamma} \cdot \frac{rh_s - rh_a}{r_a}$$

$$\therefore 1.1 \frac{e^*(T_a)}{rh_s S + \gamma} \approx \frac{e^*(T_s)}{rh_a S + \gamma} \tag{12}$$

Equation (12) reveals an emergent daily time scale relationship between temperature and relative humidity which has the potential to be used as a supplementary equation in future research.

Another important finding of the decomposition analysis is the global-scale land-atmosphere equilibrium. Our analysis in Fig. 4 (d1) indicates that the mean value of daily $LE_G$ of all FLUXNET2015 sites is close to zero, implying the

global mean $rh$ gradient is near zero at a daily time scale. Importantly, $LE$ is primarily determined by $LE_Q$ ($R^2 = 0.65$) instead of $LE_G$ ($R^2 = 0.18$) as depicted in Fig 4 (b1) and (b2). Nevertheless, FLUXNET2015 data also suggests that $LE_G$ is the main driver of local-scale variability of EF at the daily time scale (Fig. 4 (c1) and (c2)). Although the mean value of daily EF is close to the mean value of $\frac{LE_Q}{Q}$, the variation of daily EF depends more on the variation of $\frac{LE_G}{Q}$ (Fig. 4 (d2)). It should be noted that Fig. 4 and Fig. S2 are almost identical (Fig. S2 repeats the presentation shown in Fig. 4 using value computed when forcing

energy balance closure), implying that the lack of surface energy balance closure for EC observations does not significantly impact our analyses and interpretations.

### 4.3. Decomposition analysis of FLUXCOM dataset

We then applied the PM$_{rh}$ model to the FLUXCOM dataset, a benchmark global $LE$ data product (Jung et al., 2019). As shown in Fig. 5 (a) and (c), the spatial patterns of the annual mean $LE$ and $LE_Q$' were similar. For instance, both $LE$ and $LE_Q$' show

the highest values around the equator at annual time scale, which is mainly due to the energy available in this region. Also, spatial variability of $LE$ is mostly determined by $LE_Q$' ($R^2 = 0.85$ and slope = 1) rather than by $LE_G$' ($R^2 = 0.18$) (Fig 5. (f1) and (f2)). This result is consistent with Eq. (8) and the SFE theory. In other words, the land surface is generally under thermodynamic equilibrium with the atmosphere at the global-annual scale (i.e., $rh_s \approx rh_a$). Furthermore, the monthly time series of global $LE$ and its two components in Fig. 5 (e1) show that (i) $LE_G$' is consistently close to zero at the global scale,

and (ii) the seasonal variability of global $LE$ is primarily determined by $LE_Q$'.

However, while mean annual $LE_G$' was close to zero in broad areas (particularly in high latitude regions) as exemplified in Fig. 5 (e2), it was distinctly positive or negative at the annual scale for many regions (Fig. 5 (d)). In humid tropical regions like the Amazon basin where moisture convergence is large, $LE_G$' was generally positive, whereas arid regions such as Australia were characterized by negative $LE_G$' (Fig. 5 (e3) and (e4)). Here, positive $LE_G$' (i.e., $rh_s > rh_a$) indicates the

land surface is wetter than the near-surface atmosphere while negative $LE_G$' (i.e., $rh_s < rh_a$) implies a drier land surface than the atmosphere. Therefore, the sign of $LE_G$' in Fig. 5 (d) can be interpreted as representing land surface dryness relative to the atmosphere at an annual time scale.

The spatial pattern of $LE_G$' is similar to the spatial pattern of EF, but differs from the spatial pattern of $LE_Q$' (Fig. 5 (b)). For example, EF was high in not only tropical regions but also temperate climates such as Mediterranean regions, and

this spatial pattern is well matched with the spatial pattern of $LE_G$' but not $LE_Q$'. The finding that that the spatial variation of EF is primarily controlled by $LE_G$' instead of $LE_Q$' was supported by correlation analyses in Fig. 5 (f3) and (f4) ($R^2 = 0.60$ for EF~$\frac{LE_G'}{Q}$ and $R^2 = 0.28$ for EF~$\frac{LE_Q'}{Q}$). This is understandable in that EF is a reflection of the land surface dryness (Gentine et al., 2011), and $LE_G$' reveals the land surface dryness relative to the atmosphere.

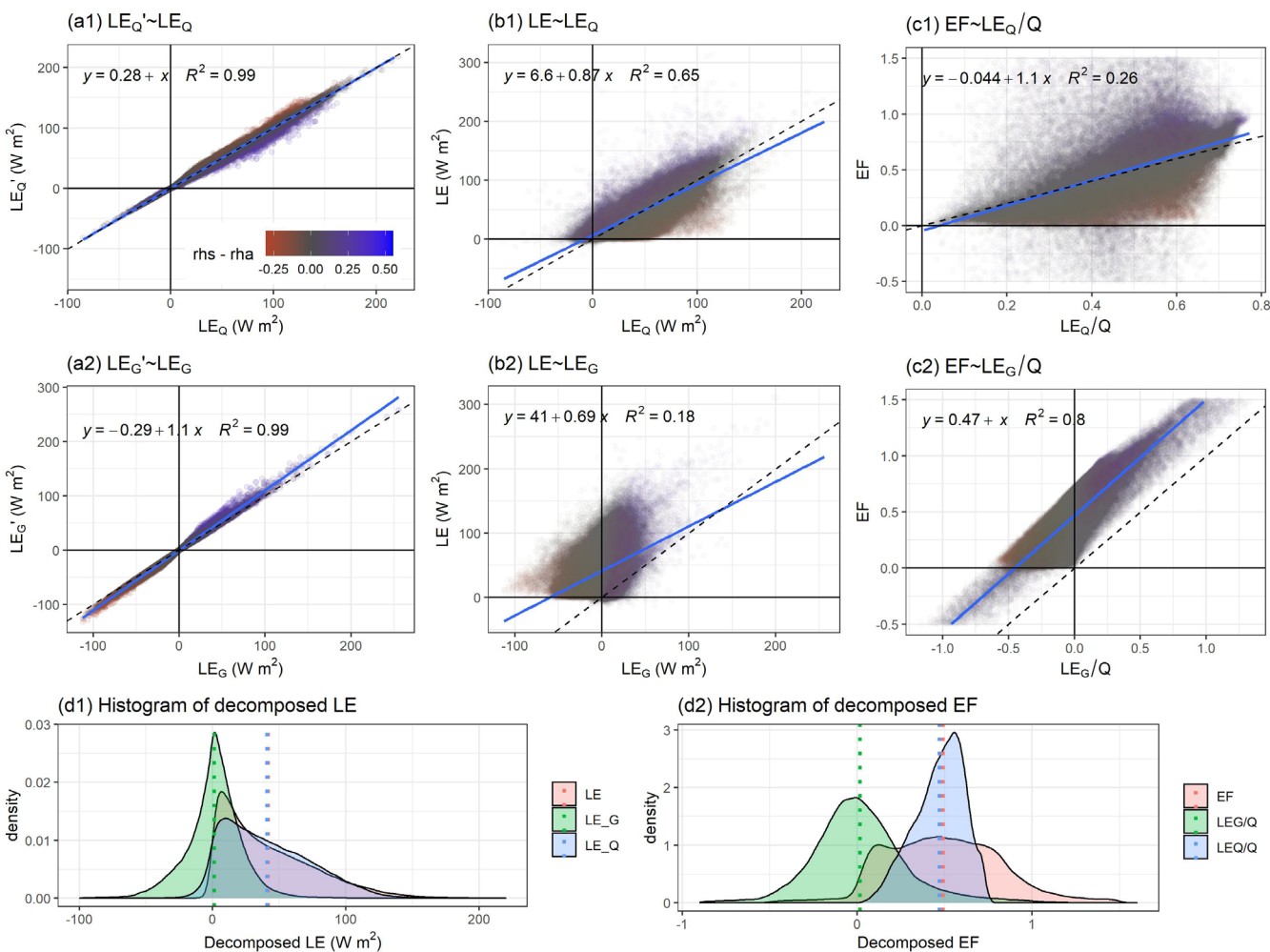


**Figure 4: FLUXNET2015 daily scale decomposed *LE* for 212 sites and 1532 site-years. Panels (a1) and (a2) are linear regressions of *LE_Q*' on *LE_Q* and *LE_G*' on *LE_G*. Panels (b1) and (b2) are linear regressions of *LE* on *LE_Q* and *LE* on *LE_G*. Panels (c1) and (c2) are linear regressions of EF on *LE_Q/Q* and EF on *LE_G/Q*. In these panels, daily EF data within a range from -1 to 1.5 are only shown. Here, dashed lines are one-to-one lines, blue lines are regression lines, and color represents *rh_s−rh_a*. Panel (d1) and (d2) are**
**histograms of decomposed *LE* and EF with mean values (dotted lines). To correct for lack of energy balance closure, *Q* was set equal to *LE+H* in all calculations.**

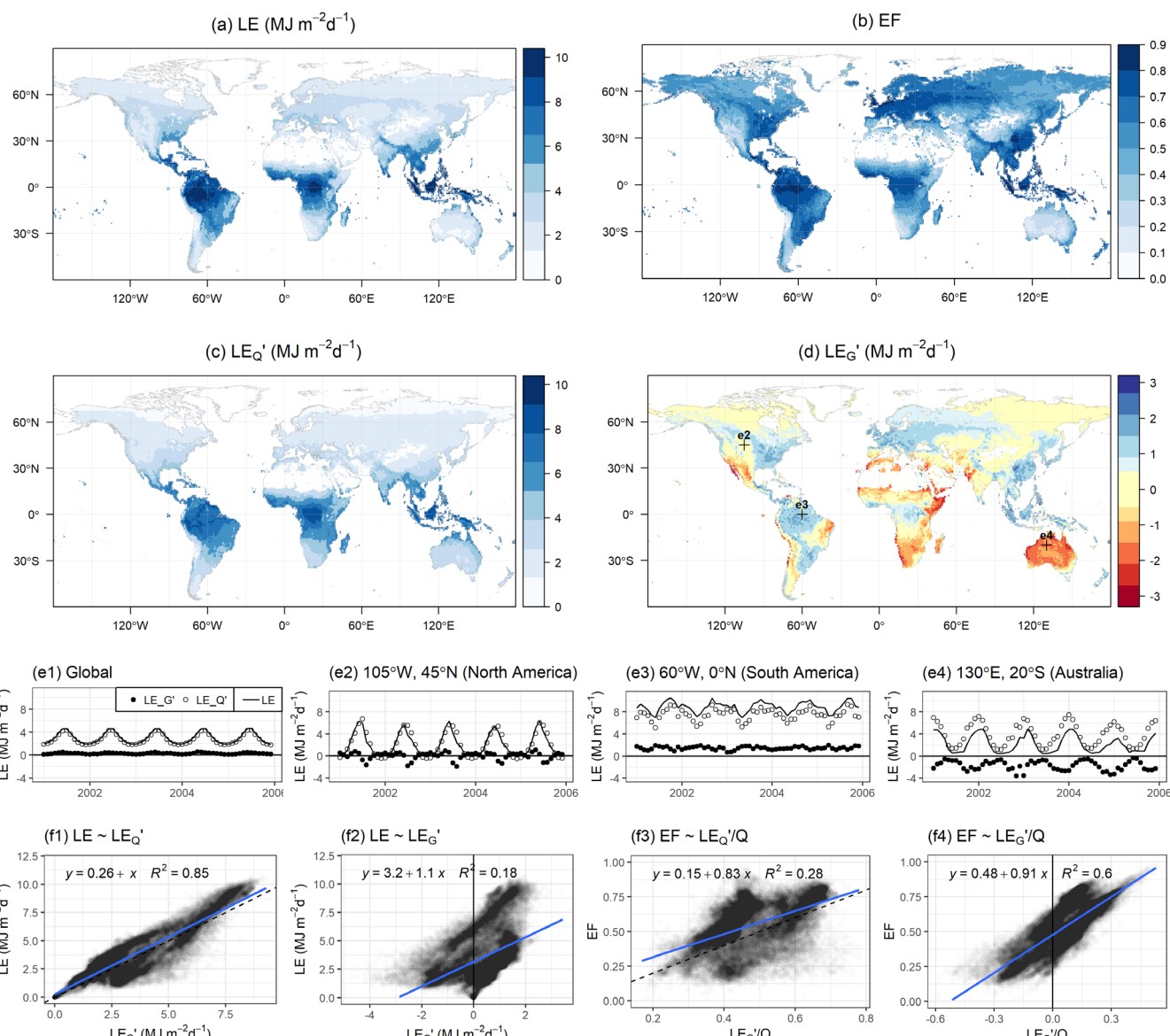

**Figure 5: Mean annual *LE*, EF, *LE_Q'*, and *LE_G'* from 2001 to 2005 (panels (a), (b), (c), and (d), respectively). Panel (e1) is a time series of monthly global average *LE* and the two components, *LE_G'* and *LE_Q'*. Panels (e2), (e3), and (e4) are time series at specific locations highlighted in panel (d). Panels (f1), (f2), (f3), and (f4) are spatial linear regressions of *LE* on *LE_Q'*, *LE* on *LE_G'*, EF on *LE_Q'/Q*, and EF on *LE_G'/Q*, respectively.**

## 5. Discussion

### 5.1. $LE_G$ at sub-daily scale

Salvucci and Gentine (2013) found that the variance of the $rh$ gradient tends to be minimized over the course of the day. Based on this empirical finding, they developed an approach to predict $LE$ only using standard meteorological measurements, and this approach accurately predicted actual $LE$ (Rigden and Salvucci, 2015, 2017). Our PM$_{rh}$ model provides theoretical support for their approach in that $LE_G$ acts to reduce the $rh$ gradient. Indeed, the U-shape diurnal cycles of $LE_G$ in Fig. 3 (positive nighttime and negative daytime) show the direction and the magnitude of the equilibration process of $rh$ gradient at a sub-daily

scale resulting in a small gradient of $rh$ on daily average.

We found positive nighttime $LE_G$ values regardless of wet or dry conditions, except when the atmosphere was fully saturated atmosphere (Fig. 3). This result is a natural consequence since the land surface is close to saturation at night. This finding suggests that $LE_G$ is a dominant contributor to nighttime evaporation since available energy is close to zero at night. It is important since nighttime evaporation is a non-negligible component of total ET (Padrón et al., 2020).

Unlike nighttime $LE_G$ values, the direction and the magnitude of the daytime $LE_G$ are highly dependent on the land surface dryness. For example, the U-shape diurnal cycles of $LE_G$ are apparent only when the land surface is dry, which is confirmed by the negative wavelet coherence between $LE$ and $LE_G$ at the diurnal time scale in Fig. 2 (f). When the land surface is wetter than the atmosphere, $LE_G$ values were positive even in the daytime, and thus the U-shape diurnal cycles of $LE_G$ did not appear (Fig 3 (c)). The positive $LE_G$ during daytime periods may be explained as a consequence of warm and dry air entrainment at the top of the atmospheric boundary layer and/or horizontal advection of sensible heat which may reduce

atmospheric relative humidity (Baldocchi et al., 2016;de Bruin et al., 2016). Indeed, a strong entrainment effect and/or local advection of sensible heat are common phenomena for irrigated agriculture (Baldocchi et al., 2016;de Bruin and Trigo, 2019), and the irrigated sugarcane site shows that the annual mean $LE_G$ was always positive except for in 2016 when there was no application of dry season irrigation.

### 5.2. $LE_G$ at daily to annual time scales


As described in the theory section, land-atmosphere equilibrium is achieved when $LE_G$ approaches zero and thus $LE$ reduces to Eq. (8) at a time scale longer than sub-daily. The decomposed terms derived from both the empirical FLUXNET2015 and model-based FLUXCOM datasets show that the global mean for $LE_G$ is near zero, implying global-scale land-atmosphere equilibrium (Fig. 4 and Fig. 5). This result extends the SFE theory of McColl et al. (2019). Although, $LE_G$ is not always near

zero for time scales longer than sub-daily, moisture convergence and divergence at the global-scale tend to balance each other out, resulting in global-scale land-atmosphere equilibrium on longer time scales (Fig. 5 (e1)).

From a different perspective, non-zero $LE_G$ value and its sign (+ vs -) at the local scale can be understood as an indicator reflecting land surface dryness relative to the atmosphere. We found that $LE_G$ clearly distinguishes spatially wet and dry regions around the world (Fig. 5 (d)). We also found that the spatiotemporal variability of EF was largely explained by

$LE_G$ instead of $LE_Q$ (Fig 4 (c2) and Fig 5 (f4)). These results demonstrate the usefulness of $LE_G$ to quantify land surface dryness. Some previous studies introduced evaporative stress index or evaporation deficit index based on the ratio (or difference) between potential evaporation and actual evaporation (Anderson et al., 2015;Kim and Rhee, 2016;Fisher et al., 2020;Baldocchi et al., 2021), but these methods are highly dependent on the way one calculates potential evaporation. Unlike potential evaporation (which is a theoretical value), $LE_G$ is a true physical quantity, and negative $LE_G$ values directly reflect water-

limited land surface conditions. Therefore, we suggest applying our decomposition method to better quantify evaporative stress.

### 5.3. Future applications

In this study, we present the $PM_{rh}$ model and demonstrate its utility for exploratory and diagnostic purposes. However, the model has potential applications for other purposes. One possible application is to use $PM_{rh}$ to predict actual ET. Although the

original PM equation is widely used to predict evapotranspiration (e.g., Leuning et al., 2008;Mu et al., 2011;Mallick et al., 2014), their accuracy often relies on parameterised surface resistance models (Polhamus et al., 2013). Since our $PM_{rh}$ formulation does not include surface resistance, it could represent a good alternative. As shown in the results section, $LE_Q'$ can be calculated using typical meteorological data without additional surface parameters. Also, we found that $LE_Q'$ alone can be used effectively to approximate $LE$. If the remotely sensed land surface conditions are known (e.g., soil moisture and/or land

surface temperature), actual $LE$ may be more accurately predicted by incorporating $LE_G$ term. To estimate the $LE_G$ term, $rh_s$ may be estimated based on soil moisture and/or land surface temperature data. For example, Eq. (12), which is well supported by observations (Fig. 4 (a2)) and on a physical basis, may be used to calculate $rh_s$.

      Another possible application is to study impacts of climate change and land use and land cover change on surface energy partitioning and evaporation. Changes in the atmospheric state such as temperature and relative humidity, as well as

changes in the land surface characteristics such as albedo and aerodynamic roughness, can alter evaporation (Lee et al., 2011;Wang et al., 2018). However, how these changes affect terrestrial energy partitioning and ET is still unclear. Indeed, there is a large discrepancy in long-term $LE$ trends among current land surface models (Pan et al., 2020). Since $PM_{rh}$ makes it possible to physically decompose $LE$ into adiabatic and diabatic thermodynamic components, the $PM_{rh}$ approach can be useful to understand how environmental changes affect surface energy partitioning. For instance, trend analysis for the decomposed

$LE$ using $PM_{rh}$ could contribute to improve our understanding of earth's climate system and water cycle.

### 5.4. Potential caveats

Despite the insights it can offer, the $PM_{rh}$ model shares several limitations with the traditional PM model. First, PM style equations linearize the exponential relationship between saturation vapor pressure and temperature (Clausius-Clapeyron relation), but the linearization can cause bias when the temperature difference between surface and atmosphere is substantial

(Paw U and Gao, 1988;McColl, 2020). Second, the $PM_{rh}$ model assumes that aerodynamic resistance for heat and water vapor are identical, which implicitly relies on the assumption that the ratio of the turbulent Schmidt to Prandtl numbers is unity

(Knauer et al., 2018a). This similarity assumption cannot be held in some cases, especially when advective flux divergence is significant (Lee et al., 2004). The third potential caveat is that we define the land surface as an idealized single plane that is equivalent to the "bigleaf" representation of the traditional PM equation, but this approach ignores profiles of temperature and
humidity inside the canopy (Bonan et al., 2021).

Another potential caveat concerns the surface energy balance. The surface energy balance is a governing equation of the PM style models, but it is not satisfied in typical EC observations, which is referred to as the "surface energy balance closure problem" (Wilson et al., 2002). This closure problem could cause a systematic uncertainty in estimating $r_s$ when using the PM equation (Knauer et al., 2018b;Wohlfahrt et al., 2009), and this issue may affect the diagnostic analyses using the $PM_{rh}$.
In this study, we did not force the energy balance closure and attributed the cause of observed surface energy imbalances to unmeasured heat storage terms for the Costa Rica EC site due to possible significant role of heat storage term (details in the section 3.1). Wehr and Saleska (2021) recently demonstrated that regardless of whether the lack of energy balance closure of EC observations is due to $LE + H$ or is due to $R_n - G$, applying the flux gradient equation to observed $LE$ and $H$ without energy balance correction is the best way in determining $r_s$. This is because applying the flux gradient equation to observed $LE$ and $H$
can dispense with the unnecessary assumption of energy balance closure. They showed that bias introduced by underestimated $LE$ and $H$ is smaller than the bias introduced by the energy balance closure assumption. This finding may be applied to our analysis in calculating $rh_s$ instead of $r_s$.

As for the FLUXNET2015 dataset, we provide an alternate analysis using energy balance corrected $LE$ and $H$ (Bowen ratio preserving method in Pastorello et al. (2020)) in the supplement. We found that the results for corrected and uncorrected
versions were almost identical, which can be viewed as a natural consequence since in Eq. (10), $LE$ and $H$ are included in the numerator and denominator respectively. Multiplying the same ratio to $LE$ and $H$ in Eq. (10) to correct $LE$ and $H$ based on the Bowen ratio method does not significantly change the resulting $rh_s$. Therefore, the lack of surface energy balance closure does not significantly impact our analyses and interpretations unless the lack of energy balance is dominated by $LE$ only or $H$ only. If the lack of energy balance is dominated by $LE$ only or $H$ only, our results and interpretation may include systematic bias.
Finally, there are several ways to calculate aerodynamic resistance, and the chosen form for $r_a$ may affect the results. However, the influence of this choice is expected to be marginal compared to the energy balance problem. Knauer et al. (2018b) showed that uncertainty caused by different $r_a$ on surface conductance is low compared to the energy balance closure problem. This finding can be applied to our analysis. Specifically, in Eq. (10), $r_a$ is multiplied by both denominator and numerator, and thus a small difference in $r_a$ should not significantly affect the resulting $rh_s$.

**6. Conclusions**

We have shown that our novel $PM_{rh}$ model provides a new opportunity to understand the governing physics of the terrestrial energy budget. Specifically, the $PM_{rh}$ model helps to illustrate how the land surface conditions become encoded to the atmospheric state by partitioning $LE$ into two thermodynamic processes. "Dry land-dry air" or "wet land-wet air" conditions

can each lead to daily scale land-atmosphere equilibrium although the diurnal pattern of the equilibration process (i.e., $LE_G$) is starkly different. Our findings suggest that while $LE_G$ is a primary component determining EF, spatiotemporal variability of $LE_Q$ alone can adequately represent the variability of $LE$. We found global-scale land-atmosphere equilibrium at daily to annual scales, which implies that global $LE$ can be simply determined by the atmospheric state and radiative energy without any surface constraint required to represent spatial heterogeneity and physiological influences. From a different perspective, non-zero $LE_G$ value at a local scale can be understood as an indicator revealing land surface dryness. Questions remain regarding how $LE_Q$ and $LE_G$ will be influenced in relation to changing climatic and land surface conditions, and how these changes might affect the climate system at differing spatial and temporal scales through positive or negative feedbacks.

*Data availability.* The FLUXNET2015 dataset is available in https://fluxnet.org/data/fluxnet2015-dataset/. The highlighted sugarcane eddy covariance site dataset will be available in AmeriFlux (https://ameriflux.lbl.gov/). The FLUXCOM dataset is available in http://www.fluxcom.org/.

*Author contribution.* Y.K. and M.S.J. designed research; U.W. provided FLUXCOM data; Y.K., L.M., and M.S.J. performed research; Y.K. analyzed data; Y.K., M.G., T.A.B, L.M., and M.S.J. wrote the paper.

*Competing interests.* The authors declare no conflict of interest.

*Acknowledgements.* We want to thank Dr. Iain Hawthorne, Pável Bautista, Dr. Silja Hund, Cameron Webster, Gretel Rojas Hernandez, Guillermo Duran Sanabria, Dr. Andrea Suarez Serrano, Dr. Ana Maria Duran, Martin Martinez, and Dr. Fermín Subirós Ruiz for field and logistical support. We also thank Dr. Martin Jung, the principal investigator of the FLUXCOM dataset. The authors would like to thank the EU and NSERC for funding, in the frame of the collaborative international Consortium AgWIT financed under the ERA-NET WaterWorks2015 Cofunded Call. This ERA-NET is an integral part of the 2016 Joint Activities developed by the Water Challenges for a Changing World Joint Programme Initiative (Water JPI).

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
