# Peer review of "Relative humidity gradients as a key constraint on terrestrial water and energy fluxes"

_Hydrology and Earth System Sciences, 2020_

## Referee Comment (RC1) · Anonymous Referee #1 · 19 Jan 2021

This interesting paper explores a formulation of ET fully independent of the surface resistance, relying instead on a humidity resistance depending on the gradient of moisture between the surface and screen level, following the ideas displayed in Monteith (1981). Conceptually this approach allows to skip any explicit dependency of the characteristics of the surface, particularly those of the vegetation. The approach is worth exploring and this manuscript intends to show us to what point it can be used, especially for interpretation of the processes in place, more than in a parameterisation mode.

In a set of recent papers McColl and colleagues showed that, for scales of a day or longer, ET is essentially determined by the balance between surface moistening and surface heating, with the consequence at these temporal scales that more atmospheric

moisture is the result of more ET, and the term describing it is equivalent to the diabatic term in the Penman-Monteith equation, therefore independent of the characteristics of the surface.

In this manuscript, the authors place themselves in this framework and try to extend it to sub-daily scale by using the humidity resistance in the adiabatic term, avoiding to prescribe any characteristic of the surface (vegetated or not). Their theoretical reasoning is easy to follow, substituting the pressure vapour by the relative humidity, and it is intended to be valid even for non-saturated surfaces through the prescription of a "surface relative humidity", rhs.

The authors do not explicitly comment or define rhs, which is still the missing piece in most of the approaches dealing with non-saturated surfaces. In the present case, they go around the problem of defining or calculating rhs by deriving a value for it by using observed LE and H values from EC-systems, using their formulation (2).

To do it they assume that the available energy Q is LE+H instead of Rn-G trying to circumvent the unavoidable problem of the closure of the surface energy budget. This decision could be understandable but it is poorly justified and the consequences of it are not reflected upon.

Unfortunately the result of these strong hypotheses concerning rhs and a discussion of the values obtained is not explicitly shown or made.

The proposed method cannot be used as an independent way to determine LE, because LE is used to determine rhs, so obtaining again LE from this rhs value would be of no use (unless I miss something). However the method is useful to separate the observed LE into its diabatic and adiabatic parts (taking into account that Q=H+LE is a tough assumption), and this is where most of the interpretation effort is put on.

When analyzing data from a site in Costa Rica it is shown that for daily values (and larger temporal scales) the diabatic part explains most of the total LE, the adiabatic

part being most significant in dry and low-vegetated conditions. At the sub-daily scale, the adiabatic part has a opposite behaviour to the diabatic one, which I interpret as turbulence tending to bring vapour towards the surface layer in the daytime, and removing it in the nighttime and early morning.

Fig 4 contains a lot of information in its 8 sub-figures, which are not really commented. A similar comment can be made about Fig 5 and its 12 sub-figures. What is the use of displaying so much information if then it is not discussed? In general section 4 would need to be more developed in terms of interpretation of results, which is now very shallow, especially sections 4.2 and 4.3.

As the paper is now sections 5 "Discussion" and 6 "Conclusions", both very short, could be merged into one larger Conclusions section. Instead a real "Discussion" section could come from an expanded version of the analysis of the results.

So I suggest to the authors that they i) clarify somehow the aims of their research in the initial parts of the paper, especially in the abstract, when one may get the impression that a new ET parameterization is presented, while in reality what we have is a nice method of analysis of the diabatic and adiabatic components of ET; ii) elaborate on the meaning of rhs; iii) explain better what are the expected consequences of their hypotheses in the ulterior data analysis (such as imposing Q=H+LE or the chosen form for ra); iv) expand they interpretation of data, currently very shallow, into a remade Discussion section; v) joint the current "Discussion" and "Conclusions" sections into a more comprehensive and developed new "Conclusions" section; vi) consider to summarise the information in the supplementary material and incorporate it straight into the manuscript

---

## Referee Comment (RC2) · Anonymous Referee #2 · 15 Feb 2021

Recommendation: major revision or reject with encouragement to resubmit

This manuscript deals with the role of relative humidity in evaporation estimates. It presents a novel approach to evaluate this role, and as such, I think this is very valuable. However, there are a couple of issues, some of which are major, that need to be addressed. I also had a hard time to follow the paper, so I think the authors need to work on a better structure. Also, the authors do not discuss any shortcomings of their study, showing a lack of critically assessing their own findings. There are certainly quite a few, as shown below, and these need to be assessed and discussed before any conclusions can be drawn from the analysis. As some of the major issues described below are likely to substantially change this manuscript in order to be addressed, at least major revisions need to be made, or, alternatively, the manuscript could be rejected with encouragement to resubmit, so that it would again enter an open review discussion.

Major comments:

1. "Land-atmosphere equilibrium". The authors describe that the relative humidity difference between the surface and the atmosphere is a key driver for evaporation, and its depletion is an indication of thermodynamic equilibrium. This picture is incorrect. The water vapor transport from the surface into the atmosphere is driven primarily by buoyancy, that is, by the sensible heat flux, not by the difference in vapor pressure. Hence, a depletion of the relative humidity difference is solely a reflection of well-mixed air near the surface. This misconception is reflected in much of the manuscript, with buoyant mixing not mentioned anywhere, so this needs to be addressed in a revision.

2. Shortcomings of the PM equation. The authors use the Penman Monteith (PM) equation as the basis of their work, although this equation has some clear deficiencies. This was, for instance, clearly shown in the paper by Milly and Dunne (2016, "Potential evapotranspiration and continental drying", Nature Climate Change), also by Renner et al. (2019, "Using phase lags to evaluate model biases in simulating the diurnal cycle of evapotranspiration: a case study in Luxembourg", HESS). The authors take the PM equation for granted, but I think they need to be more critical and evaluate potential flaws in their work in light of these shortcomings.

3. Role of advection. Literature by de Bruin (e.g., de Bruin et al., 2016, "A Thermodynamically Based Model for Actual Evapotranspiration of an Extensive Grass Field Close to FAO Reference, Suitable for Remote Sensing Application", J. Hydromet.) shows that it is basically the equilibrium evaporation term that dominates evaporation rates, and that the second term comes from advection, e.g. in the case of evaporation from irrigated land in an arid region. Given that the evaluation includes data from Costa Rica from an irrigated site during the dry season, advection could play quite an important role in shaping the evaporation estimate. But as far as I can tell, advection is not being

mentioned as a phenomenon in the manuscript. In addition to the shortcomings of the PM equation, I think there is a good reason to doubt the analysis so these factors need to be assessed.

4. Closure of the energy balance. The authors write on L177 that they did not enforce an energy balance closure, attributing the lack of closure due to unmeasured heat storage terms (but without quantifying and supporting this attribution). However, I do not think that this is a feasible way to do this analysis. It seems to me that the scientific consensus is that the imbalance is mostly attributable to secondary circulations in the convective boundary layer (see, e.g., the review by Mauder et al., 2020, Boundary Layer Meteorology, https://doi.org/10.1007/s10546-020-00529-6 ). I think this aspect also needs to be addressed thoroughly in the analysis.

Minor comments:

Title: "relative humidity gradients" are not a constraint. They are highly dependent on moisture and heating of air, hence not an independent variable, and they are not a physical constraint, such as those imposed by the energy- or water balance, or the laws of thermodynamics.

L34: "LE predictions remain highly uncertain" - I doubt this statement. The basic constraints for evaporation have been quite well established over decades, so what are the factors that remain uncertain? The authors need to be more specific than this wide-sweeping claim.

L38: Actually, the "pioneering work" on governing physics of LE started with Schmidt (1915), as described by de Bruin et al (reference provided above).

L46: "rh budget" - there is no rh budget, because relative humidity is not a conserved quantity, as it jointly depends on temperature and moisture content. So you can talk about an energy budget or a moisture budget, but not of a rh budget.

L51: I would not describe the PM equation as reflecting governing physics, it is at best

semi-empirical, with shortcomings (see above, major comment 2).

L69-83: I do not understand the derivation (and it seems odd to have this derivation partly in the introduction and partly in the appendix). What does surface relative humidity stand for? When we deal with a vegetated surface that experiences water limitation, then any water that does evaporate either diffuses out of the soil (for which I would think a surface resistance would be rather critical to consider) or it is evaporated inside a leaf, but with the exchange with the atmosphere constrained by stomatal conductance. So how do these equations (2) and (3) reflect a physical picture of the evaporation process?

Eq (4): As pointed out in the major comments, the second term in the PM equation is likely reflecting advective conditions (see paper by de Bruin et al., mentioned earlier).

L103: "When the vertical gradient of rh dissipates...": see Major Comment 1.

L114: Again, I do not understand the reasoning of the authors (see comment above, L69-83).

Figure 1: Why is the initial state represented by the atmospheric conditions, and the final state represented by the surface conditions? Doesn't evaporation start at the surface? Actually, evaporation may already start within the soil (bare soil evaporation) or inside leaves. How does this fit into this diagram? I find this quite confusing.

L120: "In the equilibrating process" - which process do you mean?

L120: "the air parcel is adiabatically cooled" - why is it cooled/lifted? Isn't it rather by the mixing of the moistened air from the surface with the unsaturated air of the atmosphere due to buoyancy that depletes the difference?

L149: "rh budget" - relative humidity is not a conserved quantity, so there is no budgeting. If you talk about mass conservation, you would need to formulate this in terms of specific humidity or similar.

[Figure]

L151: "This is logical in that LE$_G$ itself operates to diminish the vertical rh gradient." No, it does not. Vertical mixing is related to buoyancy, not to VPD. Although mixing also depletes the rh difference, it is not the same process.

L152/53: The classical equilibrium evaporation rate $LE = s/(s + \gamma) * Q$ is not derived from the assumption of a saturated atmosphere. In the original derivation by Schmidt (1915), the only assumption is that the air immediately in contact with an open water surface is in thermodynamic equilibrium, so that the addition of energy is partitioned accordingly. But it does not assume that the atmosphere is saturated.

L194-201: I don't understand the need for the wavelet analysis. Why is it necessary? It seems to me that it makes the analysis more complicated than necessary.

L203-212: I am skeptical about the use of daily averages. Relative humidity, wind speeds, air and surface temperatures, and aerodynamic conductance show pronounced variations at the diurnal scale. How do you account for the covariations among these variables if you use daily means?

L220-226: Same question: How do you account for covariations among variables when you use daily mean forcing?

L229-234: I do not understand what the wavelet analysis should tell me. Why do you not simply use autocorrelations?

L240: That $LE_G$ is close to zero in the absence of irrigation in 2016 supports the interpretation mentioned above that the second term in the PM equation relates to an advection effect.

L245: Figure 2: I would appreciate a little more information of the site - like precipitation input, solar radiation etc. to provide more background about the site.

L256: Does the case shown in Figure 3c represent a case with irrigation? Then, I guess, $LE_G$ relates to advection effects?

[Figure]

L265: Figure 3: I would find it informative to also see H and net radiation, as well as the diurnal variations in the rh's.

L286: The statement that "the land surface is generally under thermodynamic equilibrium with the atmosphere at the global-annual scale" is, I think, incorrect. The finding that rhs ≈ rha simply means that the air near the surface is well mixed, likely due to buoyancy.

L322: "land-atmosphere equilibrium is achieved ..." - again, the authors neglect the role of buoyancy that mixes the air near the surface here and which is likely to play the dominant role in reducing the relative humidity difference.

L329-334: I think the implications need to be rethought, given that the role of buoyancy in depleting a difference in rh has been neglected.

L335: Conclusions - same here, given the methodological flaws of the study, this paragraph needs to be rethought.

―――――――――――

---

## Author Comment (AC1) · 14 Mar 2021

**Response to anonymous referee #1**

**Relative humidity gradients as a key constraint on terrestrial water and energy fluxes (HESS-2020-643)**

March 14, 2021

Dear Anonymous Referee #1,

First of all, thank you very much for reviewing our manuscript.

We were pleased to receive your comments and suggestions. In the following, we present our detailed responses to your suggestions with our replies in blue; the specific revisions that we intend to perform are indicated by underlined text. In this response, we reconstructed your comments based on the last paragraph's suggestions while covering all of the comments.

On behalf of all authors,

With regards,

Yeonuk Kim,

Corresponding author

**1. General comment**

This interesting paper explores a formulation of ET fully independent of the surface resistance, relying instead on a humidity resistance depending on the gradient of moisture between the surface and screen level, following the ideas displayed in Monteith (1981). Conceptually this approach allows to skip any explicit dependency of the characteristics of the surface, particularly those of the vegetation. The approach is worth exploring and this manuscript intends to show us to what point it can be used, especially for interpretation of the processes in place, more than in a parameterisation mode.

In a set of recent papers McColl and colleagues showed that, for scales of a day or longer, ET is essentially determined by the balance between surface moistening and surface heating, with the consequence at these temporal scales that more atmospheric moisture is the result of more ET, and the term describing it is equivalent to the diabatic term in the Penman-Monteith equation, therefore independent of the characteristics of the surface.

In this manuscript, the authors place themselves in this framework and try to extend it to sub-daily scale by using the humidity resistance in the adiabatic term, avoiding to prescribe any characteristic of the surface (vegetated or not). Their theoretical reasoning is easy to follow, substituting the pressure vapour by the relative humidity, and it is intended to be valid even for non-saturated surfaces through the prescription of a "surface relative humidity", $rh_s$.

We thank Reviewer #1 for your interest and constructive feedback on the manuscript. Your summary is consistent with what we wish to communicate regarding our study.

**2. Specific suggestion 1**

*Suggestions:* clarify somehow the aims of their research in the initial parts of the paper, especially in the abstract, when one may get the impression that a new ET parameterization is presented, while in reality what we have is a nice method of analysis of the diabatic and adiabatic components of ET;

*Critique:* The proposed method cannot be used as an independent way to determine $LE$, because $LE$ is used to determine $rh_s$, so obtaining again $LE$ from this $rh_s$ value would be of no use (unless I miss something). However the method is useful to separate the observed $LE$ into its diabatic and adiabatic parts (taking into account that $Q=H+LE$ is a tough assumption), and this is where most of the interpretation effort is put on.

Thank you for your suggestion. We agree with your suggestion. We focused on diagnostic analyses instead of predictive analysis using the proposed PM$_{rh}$ model in order to demonstrate the importance of partitioning $LE$ into diabatic and adiabatic components to improve understanding of the surface energy balance. Therefore, we will follow your suggestion by revising the abstract to explicitly highlight the decomposition analysis.

In this paper, we applied the new PM$_{rh}$ model for a diagnostic purpose (i.e., using the model to aid in interpretation of governing mechanisms of measured or predicted ET). As such, $rh_s$ was determined by measured $LE$ and $H$ in this study. However, it should be noted that the proposed PM$_{rh}$ model has the potential to be used to determine ET in principle if a model to determine $rh_s$ is available, although it is beyond the scope of the study. For instance, a combination of surface temperature and soil moisture (remotely sensed or field measurement) can be used to predict $rh_s$ (e.g., Hajji et al., 2018). We will describe this point in the discussion section to suggest potential applications of the proposed model in the future.

**3. Specific suggestion 2**

*Suggestions:* elaborate on the meaning of $rh_s$;

*Critique:* The authors do not explicitly comment or define $rh_s$, which is still the missing piece in most of the approaches dealing with non-saturated surfaces. In the present case, they go around the problem of defining or calculating $rh_s$ by deriving a value for it by using observed *LE* and *H* values from EC-systems, using their formulation (2).

As described in the manuscript (L77), $rh_s$ is the relative humidity at the land surface, and is a purely physical quantity (vapour pressure divided by saturation vapour pressure). Here, the land surface can be defined as a single plane located at $d+z_{0h}$ (d=displacement height, $z_{0h}$ = roughness length for heat) following the bigleaf framework of micrometeorology (Knauer et al., 2018a). We noticed that the land surface was not explicitly defined in the preprint version, and thus we will add the definition of $rh_s$ and the land surface explicitly in the theory section.

**4. Specific suggestion 3**
*Suggestions:* explain better what are the expected consequences of their hypotheses in the ulterior data analysis (such as imposing *Q=H+LE* or the chosen form for $r_a$);

*Critique:* To do it they assume that the available energy *Q* is *LE+H* instead of $R_n$-G trying to circumvent the unavoidable problem of the closure of the surface energy budget. This decision could be understandable but it is poorly justified and the consequences of it are not reflected upon. Unfortunately the result of these strong hypotheses concerning $rh_s$ and a discussion of the values obtained is not explicitly shown or made.

Thank you for pointing out this problem. We agree that the well-known surface energy balance closure problem for eddy covariance (EC) observations ($R_n - G > H+LE$) can cause systematic uncertainty in our analysis. For instance, $rh_s$ can be underestimated if we use $H = R_n - G - LE$ in equation (8) instead of observed *H* using EC system (L190). We appreciate your concern and we will add discussion points regarding the energy balance closure problem and chosen form for aerodynamic resistance ($r_a$). A more thorough examination of this point is presented below in this response to Reviewer comments.

As described in L172-179, we determined that most of the energy imbalance in our site is contributed by unmeasured canopy and soil heat storages. Although we cannot exactly quantify these storage terms, we reason that this is the primary source of energy imbalance as follows: First, it is expected that unmeasured canopy and soil heat storages in this site are significant since the sugarcane canopy grows up to 3.6 m tall with a dense canopy. Indeed, when canopy height was less than 1 m, the surface energy balance was very nearly closed (97%), whereas the closure was 83 % when canopy height was higher than 1 m (see L174~175).  This result supports our reasoning concerning canopy storage terms. Also, it is widely accepted that the influence of secondary circulations on the energy balance closure is small for a homogenous landscape (Mauder et al., 2020;Stoy et al., 2013;Leuning et al., 2012). Since our site is located within a homogenous landscape (Figure R1), the influence of secondary circulations on the energy balance closure may be negligible. Even if the lack of energy balance is due to underestimation of *LE + H*, there is no consensus on a universally appropriate method to correct *LE* and *H* (Mauder et al., 2020). Therefore, we did not force energy balance closure for the Costa Rica site.

Wehr and Saleska (2021) recently demonstrated that regardless of whether the lack of energy balance of EC observation is due to *LE + H* or due to $R_n - G$, applying the flux gradient equation to the observed *LE* and *H* without applying an energy balance correction is the best approach to determining surface resistance (conductance). This is because applying the flux gradient equation to the observed *LE* and *H* dispenses with the unnecessary assumption of energy balance closure (i.e., $LE + H = R_n - G$). They showed that bias introduced by underestimated *LE* and *H* is smaller than the bias introduced by the energy balance closure assumption. This finding may be applied to our analysis, although in our case we are

calculating $rh_s$ instead of surface conductance. This is another reason why we imposed $Q$ as $H+LE$ and do not enforce energy balance.

As for the FLUXNET dataset, we provided an analysis using energy balance corrected $LE$ and $H$ (Bowen ratio preserving method in Pastorello et al. (2020)) in the supplementary file; the results for corrected and uncorrected versions were almost identical. This is actually a natural consequence. In equation (8), $LE$ and $H$ are included in the numerator and denominator respectively. Multiplying the same ratio to $LE$ and $H$ in equation (8) to correct $LE$ and $H$ based on the Bowen ratio method does not significantly change the resulting $rh_s$. Therefore, the lack of surface energy balance closure does not significantly impact our analyses and interpretations unless the lack of energy balance is dominated by $LE$ only or by $H$ only. If the lack of energy balance is dominated by $LE$ only or by $H$ only, our results and interpretation could include systematic bias, representing a shortcoming of our approach. We will discuss this issue in the revised version.

As for the chosen form of $r_a$, the influence of this choice is also expected to be marginal compared to the energy balance problem. Knauer et al. (2018b) showed that uncertainty caused by different $r_a$ on surface conductance is low compared to the energy balance closure problem. This finding can be applied to our analysis. Specifically, in equation (8), $r_a$ is multiplied by both denominator and numerator, and thus a small difference in $r_a$ should not significantly affect the resulting $rh_s$.

[Figure]

*Figure R1 Costa Rica sugarcane site satellite view (retrieved from Google Earth)*

**5. Specific suggestion 4**
*Suggestions:* expand they interpretation of data, currently very shallow, into a remade Discussion section;

*Critique:* Fig 4 contains a lot of information in its 8 sub-figures, which are not really commented. A similar comment can be made about Fig 5 and its 12 sub-figures. What is the use of displaying so much information if then it is not discussed? In general section 4 would need to be more developed in terms of interpretation of results, which is now very shallow, especially sections 4.2 and 4.3.

Thank you for your suggestion. In the revised version of the manuscript, we will provide subsections of the Discussion section similar to the current Results subsections, and expand our interpretation. Also, we will more clearly explain Figures 4 and 5 in the Results section.

Specifically, we will discuss why the spatiotemporal variability of evaporative fraction is explained by $LE_G$ (adiabatic term) instead of $LE_Q$ (diabatic term). Also, we will compare diurnal and seasonal variability of $LE_Q$ and $LE_G$ terms in Figure 3 and Figure 5 (e1)~(e4). The spatiotemporal variability of $LE_G$ in Figure 5 will be interpreted further by considering possible connections between the $LE_G$ term and precipitation.

**6. Specific suggestion 5**
*Suggestions:* joint the current "Discussion" and "Conclusions" sections into a more comprehensive and developed new "Conclusions" section;

*Critique:* As the paper is now sections 5 "Discussion" and 6 "Conclusions", both very short, could be merged into one larger Conclusions section. Instead a real "Discussion" section could come from an expanded version of the analysis of the results.

Thank you for your suggestion. We will follow your suggestion by rewriting the Discussion section as mentioned above and transferring a part of the current Discussion into the Conclusion.

**7. Specific suggestion 6**
*Suggestions:* consider to summarise the information in the supplementary material and incorporate it straight into the manuscript

We felt that providing detailed information on the field observations is valuable for this paper, and thus we prepared the supplementary material. The reason we structured the presentation as the main paper plus a supplementary file is that the manuscript already covers a lot of information (e.g., new theory, and results from three different datasets), and incorporating supplementary material directly into the manuscript may overwhelm some readers. We will endeavor to revise the manuscript as concisely as possible while including essential information by expanding the Results and Discussion sections.

**References**
Hajji, I., Nadeau, D. F., Music, B., Anctil, F., and Wang, J.: Application of the Maximum Entropy Production Model of Evapotranspiration over Partially Vegetated Water-Limited Land Surfaces, Journal of Hydrometeorology, 19, 989-1005, 10.1175/jhm-d-17-0133.1, 2018.

[revised manuscript text omitted]

---

## Author Comment (AC2) · 14 Mar 2021

**Response to anonymous referee #2**

**Relative humidity gradients as a key constraint on terrestrial water and energy fluxes (HESS-2020-643)**

March 14, 2021
Dear Anonymous Referee #2,

First of all, thank you very much for reviewing our manuscript.

We are pleased to receive your comments and suggestions. In the following, we present our detailed responses to your suggestions with our replies in blue; the specific revisions we intend to perform in underlined.

On behalf of all authors,

With regards,

Yeonuk Kim,

Corresponding author

**1. General comment**

Recommendation: major revision or reject with encouragement to resubmit

This manuscript deals with the role of relative humidity in evaporation estimates. It presents a novel approach to evaluate this role, and as such, I think this is very valuable. However, there are a couple of issues, some of which are major, that need to be addressed. I also had a hard time to follow the paper, so I think the authors need to work on a better structure. Also, the authors do not discuss any shortcomings of their study, showing a lack of critically assessing their own findings. There are certainly quite a few, as shown below, and these need to be assessed and discussed before any conclusions can be drawn from the analysis. As some of the major issues described below are likely to substantially change this manuscript in order to be addressed, at least major revisions need to be made, or, alternatively, the manuscript could be rejected with encouragement to resubmit, so that it would again enter an open review discussion.

We thank Reviewer #2 for the constructive feedback on the manuscript. We will endeavor to improve the structure of the manuscript and address the issues highlighted by the Reviewer in a revised manuscript. Below we provide point by point responses to issues raised along with how we will improve our manuscript based on your suggestions.

**2. Major comment 1**

"Land-atmosphere equilibrium". The authors describe that the relative humidity difference between the surface and the atmosphere is a key driver for evaporation, and its depletion is an indication of thermodynamic equilibrium. This picture is incorrect. The water vapor transport from the surface into the atmosphere is driven primarily by buoyancy, that is, by the sensible heat flux, not by the difference in vapor pressure. Hence, a depletion of the relative humidity difference is solely a reflection of well-mixed air near the surface. This misconception is reflected in much of the manuscript, with buoyant mixing not mentioned anywhere, so this needs to be addressed in a revision.

Thank you for pointing out the issue. In this criticism, the Reviewer implies that "equilibrium" is a concept related to atmospheric stability or generating vertical motion of a fluid. From this point of view, our "land-atmosphere equilibrium" would indeed be a misconception since it is not related to generating turbulent mixing. However, this interpretation overlooks other types of thermodynamic equilibria. Although the "equilibrium" term in this manuscript is not related to generating turbulence, it is tightly related to chemical equilibrium (Kleidon et al., 2009) (see L90~113), and steady-state equilibrium evaporation inside an atmospheric boundary layer (ABL) model (McColl et al., 2019) (see L141~155).

We agree that a depletion of the $rh$ difference reflects well-mixed air near the surface. Nevertheless, this well-mixed situation can be understood as an "equilibrium" condition without representing a misconception. From a thermodynamic point of view, the $LE_G$ term associated to the dissipation of the $rh$ vertical gradient can be seen as a process producing entropy by bringing the chemical potential of the surface water to that of the atmosphere (Kleidon et al., 2009). Also, the term "equilibrium" in boundary layer meteorology is dynamically defined as the steady-state of an ABL (McNaughton and Jarvis, 1983;Raupach, 2001;McColl et al., 2019). For instance, McColl et al. (2019) stated that *"While there are various definitions of equilibrium ET, arguably the most fundamental is the evaporative state achieved by a closed system forced with constant incoming radiation that is partitioned at the lower boundary between latent and sensible heat fluxes"*. As we addressed in L141~155, evaporation approaches a steady-state of Equation (6) in an ABL model (McColl et al., 2019). We showed that this condition is equivalent to a depletion of the $rh$ difference. Therefore, the proposed $PM_{rh}$ model can be understood as an extension of the new theory by McColl et al. (2019); we highlight this point in relation to the equilibrium concept as presented in our study.

We agree that buoyancy generated by temperature gradients enhances vertical motion on top of the mechanic turbulence increasing diffusivity of eddies transporting both heat and vapour. To account for diffusivity in the flux gradient equation, buoyancy driven vertical motion is parameterized in aerodynamic resistance ($r_a$), which includes a thermal stability correction function (see equation (7)). Our derivation of $PM_{rh}$ model stems from the flux gradient equation (see Appendix A), and thus buoyancy-driven energy is implicit in $r_a$. It should be noted that the *rh* difference between the land and the atmosphere does not affect $r_a$ in our framework. We will more explicitly describe $r_a$ in the revised manuscript along with a mention of buoyancy considerations to reduce confusion.

A key aspect to consider is that thermal gradients (and sensible heat flux) are not independent of the vapour gradients (and latent heat flux) and the *rh* gradients can help to capture these linked effects. For example, in a dry land situation (e.g. *rh* of land drier than *rh* of the atmosphere) typically the atmospheric thermal lapse rate will be negative. The $rh_s$ - $rh_a$ will be negative and the $LE_G$ acts to equilibrate the drier land. The role of the thermal gradients is implicit in the $e*$ (saturation vapour pressure): the larger the thermal gradients the more negative will be the $LE_G$ term and the effect of thermal convection or buoyancy ($r_a$ will decrease) will further enhance this. We will discuss the thermal lapse rate effect on the evapotranspiration components in the next version.

**3. Major comment 2**
Shortcomings of the PM equation. The authors use the Penman Monteith (PM) equation as the basis of their work, although this equation has some clear deficiencies. This was, for instance, clearly shown in the paper by Milly and Dunne (2016, "Potential evapotranspiration and continental drying", Nature Climate Change), also by Renner et al. (2019, "Using phase lags to evaluate model biases in simulating the diurnal cycle of evapotranspiration: a case study in Luxembourg", HESS). The authors take the PM equation for granted, but I think they need to be more critical and evaluate potential flaws in their work in light of these shortcomings.

We agree that the PM equation has several limitations. However, we did not intend to take the PM equation for granted. Rather, the proposed $PM_{rh}$ model is introduced in order not to include surface resistance in the evaporation model; surface resistance is the key parameter of the PM equation (please see L64~70). The above-referred studies (Milly and Dunne, 2016;Renner et al., 2019) showed that bias introduced by the PM equation mostly originates from inaccurate representation of surface resistance (or conductance), and our proposed $PM_{rh}$ model does not include surface resistance as a means to overcome this shortcoming in the PM equation.

Of course, the proposed $PM_{rh}$ model shares other shortcomings with the PM equation, such as the linearization of the saturation vapour pressure slope, the assumption of identical aerodynamic resistances for sensible heat and water vapour, and the assumption that the land surface can be treated as a single plane (i.e., bigleaf). We agree that these shortcomings should be acknowledged in the manuscript, and thus we will explicitly describe them in the updated version as limitations of the proposed $PM_{rh}$ model.

**4. Major comment 3**
Role of advection. Literature by de Bruin (e.g., de Bruin et al., 2016, "A Thermodynamically Based Model for Actual Evapotranspiration of an Extensive Grass Field Close to FAO Reference, Suitable for Remote Sensing Application", J. Hydromet.) shows that it is basically the equilibrium evaporation term that dominates evaporation rates, and that the second term comes from advection, e.g. in the case of evaporation from irrigated land in an arid region. Given that the evaluation includes data from Costa Rica from an irrigated site during the dry season, advection could play quite an important role in shaping the evaporation estimate. But as far as I can tell, advection is not being mentioned as a phenomenon in the manuscript. In addition to the shortcomings of the PM equation, I think there is a good reason to doubt the analysis so these factors need to be assessed.

Thank you for pointing out this important issue. Under local advection of sensible heat (i.e., horizontal sensible heat advection from an adjacent dry field), eddy covariance (EC) observations no longer represent net fluxes from a control volume (Leuning, 2004). In these conditions, the energy balance equation ($LE + H = R_n – G$) for the control volume may be rewritten as $LE + H = R_n – G + Q_{adv}$ (de Bruin et al., 2016), where $Q_{adv}$ represents the sensible heat horizontally advected.

In our sugarcane site in Costa Rica, however, there is no evidence that local advection plays an important role. The flux observation plot was surrounded by similar sugarcane agriculture which also utilizes irrigation in dry season (Figure R1). Around this homogenous landscape, effects of local advection can be marginal (Leuning et al., 2012). Also, daily mean $H$ was rarely negative in dry season regardless of irrigation application (Figure R2). Negative values of $H$ are used as an indicator of local advection (Kutikoff et al., 2019). Therefore, local advection of sensible heat may not be significant in our study site. We will add this information to the supplementary material to support our findings in the next version.

Further, even if local advection plays a non-negligible role, our decomposition approach (i.e., $LE$ into $LE_Q$ and $LE_G$) is not affected by $Q_{adv}$ in principle. Since we defined available energy ($Q$) as $LE + H$ instead of $R_n – G$ in calculating $rh_s$ (L178), the decomposition approach is simply determined by $LE$ and $H$ using the EC technique, regardless of whether $LE + H = R_n – G + Q_{adv}$ or $LE + H = R_n – G$. Therefore, even under local advection conditions, the decomposition approach is robust. However, one of the assumptions of the $PM_{rh}$ model, identical eddy diffusivities for water vapour and sensible heat, can be problematic under strong local advection conditions (Lee et al., 2004). This limitation will be discussed as a shortcoming of our approach in the next version.

If $Q_{adv}$ plays a non-negligible role, it should increase $LE_G$ term in principle as the Reviewer also argued in the minor comments (see 12. Minor comment 7, 25. Minor comment 20, 27. Minor comment 22). This is because local advection of sensible heat is typically accompanied by negative $H$ (de Bruin and Trigo, 2019), which implies a large $LE_G$ value. We agree with these points and the influence of advection on $LE_G$ term will be discussed in the revised manuscript.

[Figure]

*Figure R1 Costa Rica sugarcane site satellite view (retrieved from Google Earth)*

[Figure]

*Figure R2 Time series of daily mean H for the sugarcane EC tower site in Costa Rica. Background color is atmospheric relative humidity (rh$_a$), and thus low rh$_a$ implies dry season. Dashed line with "h" indicates sugarcane harvest.*

**5. Major comment 4**

Closure of the energy balance. The authors write on L177 that they did not enforce an energy balance closure, attributing the lack of closure due to unmeasured heat storage terms (but without quantifying and supporting this attribution). However, I do not think that this is a feasible way to do this analysis. It seems to me that the scientific consensus is that the imbalance is mostly attributable to secondary circulations in the convective boundary layer (see, e.g., the review by Mauder et al., 2020, Boundary Layer Meteorology, https://doi.org/10.1007/s10546-020-00529-6 ). I think this aspect also needs to be addressed thoroughly in the analysis.

Thank you for pointing this out. We agree that the well-known surface energy balance closure problem of eddy covariance (EC) observations ($R_n - G > H+LE$) can lead to systematic uncertainty in our analysis, which we also discussed in our response to Reviewer 1. For instance, $rh_s$ can be underestimated if we use $H = R_n - G - LE$ instead of using EC-observed $H$ in equation (8) (L190). We understand your concern, but we still believe our approach is an appropriate way to deal with this issue. We will add an in-depth discussion regarding the energy balance closure problem based on the below paragraphs.

As described in L172-179, we hypothesize that most of the energy imbalance in our site may be contributed by unmeasured canopy and soil heat storages in the soil layer between the soil heat flux plates and the surface. Although we cannot exactly quantify the storage terms, we have clear reasoning regarding the role of heat storages in our energy imbalance. First, it is expected that unmeasured canopy and soil heat storages in this site are significant since the sugarcane canopy grew up to 3.6 m tall with a dense canopy. Indeed, when the canopy height was less than 1 m, the surface energy balance was almost closed (97%), whereas the closure was 83% when canopy height was higher than 1 m (see L174~175). This result supports our reasoning. Also, it is widely accepted that the influence of secondary circulations on the energy balance closure is small for a homogenous landscape (Mauder et al., 2020;Stoy et al., 2013;Leuning et al., 2012). Since our site is located within a homogenous landscape (Figure R1), the influence of secondary circulation on the energy balance closure may be negligible. Even if the lack of energy balance is due to an underestimation of $LE + H$, there is no consensus on a universally appropriate method to correct $LE$ and $H$ (Mauder et al., 2020). Therefore, we did not enforce energy balance closure for the Costa Rica site.

Wehr and Saleska (2021) recently demonstrated that regardless of whether the lack of energy balance closure of EC observations is due to $LE + H$ or $R_n - G$, applying the flux gradient equation to observed $LE$ and $H$ without energy balance correction is the best way in determining surface resistance (conductance). This is because applying the flux gradient equation to the observed $LE$ and $H$ can dispense with the unnecessary assumption of energy balance closure (i.e., $LE + H = R_n - G$). They showed that bias

introduced by underestimated *LE* and *H* is smaller than the bias introduced by the energy balance closure assumption. This finding may be applied to our analysis in calculating $rh_s$ instead of surface conductance. For instance, as described above (4. Major comment 3), it is better to use observed *LE* and *H* to calculate $rh_s$ under local advection conditions since "$LE + H = R_n – G$" does not hold under advection conditions. This is another reason why we imposed *Q* as *H+LE* and do not enforce energy balance.

As for the FLUXNET dataset, we provided analysis using energy balance corrected *LE* and *H* (Bowen ratio preserving method in Pastorello et al. (2020)) in the supplement. We found that the results for corrected and uncorrected versions were almost identical. This is a natural consequence. In equation (8), *LE* and *H* are included in the numerator and denominator respectively. Multiplying the same ratio to *LE* and *H* in equation (8) to correct *LE* and *H* based on the Bowen ratio method does not significantly change the resulting $rh_s$. Therefore, the lack of surface energy balance closure does not significantly impact our analyses and interpretations unless the lack of energy balance is dominated by *LE* only or *H* only. If the lack of energy balance is dominated by *LE* only or *H* only, our results and interpretation may include systematic bias, and it is a shortcoming of this research. We will discuss this issue in the next version.

**6. Minor comment 1**
Title: "relative humidity gradients" are not a constraint. They are highly dependent on moisture and heating of air, hence not an independent variable, and they are not a physical constraint, such as those imposed by the energy- or water balance, or the laws of thermodynamics.

We agree that relative humidity is a physical quantity that depends on both moisture and heat of air. However, the main finding of this study and implication of the proposed model is that vertical relative humidity gradients constrain the latent heat flux (and thus evapotranspiration). This has not been recognized previously, but we feel that we have demonstrated this point in the present study.

**7. Minor comment 2**
L34: "LE predictions remain highly uncertain" - I doubt this statement. The basic constraints for evaporation have been quite well established over decades, so what are the factors that remain uncertain? The authors need to be more specific than this wide-sweeping claim.

In this sentence, we intended to highlight that spatiotemporal variability in land surface dryness makes it difficult to predict actual *LE*. Following your opinion, we will revise this sentence to be more specific.

**8. Minor comment 3**
L38: Actually, the "pioneering work" on governing physics of LE started with Schmidt (1915), as described by de Bruin et al (reference provided above).

Thank you for your suggestion. We will acknowledge Schmidt (1915) in the next version.

**9. Minor comment 4**
L46: "rh budget" - there is no rh budget, because relative humidity is not a conserved quantity, as it jointly depends on temperature and moisture content. So you can talk about an energy budget or a moisture budget, but not of a rh budget.

Thank you for pointing out the erroneous term. We will revise it in the next version. ("rh budget" → "rh changes with time")

**10. Minor comment 5**
L51: I would not describe the PM equation as reflecting governing physics, it is at best semi-empirical, with shortcomings (see above, major comment 2).

We agree that PM equation is semi-empirical although some of its physics is correct. We will revise this sentence. ("understand the governing physics of" → "express")

**11. Minor comment 6**
L69-83: I do not understand the derivation (and it seems odd to have this derivation partly in the introduction and partly in the appendix). What does surface relative humidity stand for? When we deal with a vegetated surface that experiences water limitation, then any water that does evaporate either diffuses out of the soil (for which I would think a surface resistance would be rather critical to consider) or it is evaporated inside a leaf, but with the exchange with the atmosphere constrained by stomatal conductance. So how do these equations (2) and (3) reflect a physical picture of the evaporation process?

The full derivation of the proposed $PM_{rh}$ model is in the appendix, and equations (2) and (3) in introduction are results of the derivation. There is no derivation in the introduction. We will clarify the paragraph (L69-83).

The surface relative humidity ($rh_s$) represents a physical quantity, relative humidity (i.e., vapour pressure divided by saturation vapour pressure) at the land surface. Here, the land surface is defined as a single plane located at $d+z_{0h}$ ($d$=displacement height, $z_{0h}$ = roughness length for heat) following the bigleaf framework of micrometeorology (Knauer et al., 2018). We will add the definitions of the relative humidity and the land surface explicitly in the theory section.

The proposed $PM_{rh}$ model (equations (2) and (3)) conceptually bypasses the need to characterize surface resistance in order to describe water vapour transport and it is the key novelty of the model. The $PM_{rh}$ model is not impacted by whether the water vapour flux originates from the soil or from inside stomata of vegetation. Instead, the model only describes exchange of water vapour and heat from the land surface to the atmosphere through a turbulent process, which is parameterized by $r_a$. The focus of the model is to decompose water vapour exchange into two thermodynamic processes following the notion of Monteith (1981). In this way, evaporation can be understood as a combination of diabatic and adiabatic processes.

**12. Minor comment 7**
Eq (4): As pointed out in the major comments, the second term in the PM equation is likely reflecting advective conditions (see paper by de Bruin et al., mentioned earlier).

We agree that a large resulting value for the second term in the Penman equation can be an indication of advection. We will include the role of local advection of sensible heat in the next version. However, it is worth noting that even under the advection-free conditions, the second PM term is still required. de Bruin et al. (2016) expressed wet surface (irrigated grass) evaporation under advection-free conditions as $LE = \frac{s}{s+\gamma} Q + \beta$, where, $s$ is the linearized slope of saturation vapour pressure versus temperature, $\gamma$ is the psychrometric constant, $Q$ is the available energy, and $\beta$ is a constant correction. They introduced $\beta$ due to unsaturated air inside the atmospheric boundary layer even under advection-free conditions. $\beta$ should be understood as the second term of the Penman equation in our equations (4) and (5).

**13. Minor comment 8**
L103: "When the vertical gradient of rh dissipates...": see Major Comment 1.

Please see our response above (2. Major Comment1); we do not believe this expression to be a misconception.

**14. Minor comment 9**
L114: Again, I do not understand the reasoning of the authors (see comment above, L69-83).

Please see our response above (11. Minor comment 6); the focus of the proposed PM$_{rh}$ model is to decompose water vapour exchange into two thermodynamic processes following the notion of Monteith (1981). In this section of our manuscript, we interpreted the PM$_{rh}$ model using the psychrometric chart. Interpretation of evaporation as two thermodynamic processes using the psychrometric chart is a widely accepted approach (Monteith, 1981;Monteith and Unsworth, 2013;Monson and Baldocchi, 2014).

**15.Minor comment 10**
Figure 1: Why is the initial state represented by the atmospheric conditions, and the final state represented by the surface conditions? Doesn't evaporation start at the surface? Actually, evaporation may already start within the soil (bare soil evaporation) or inside leaves. How does this fit into this diagram? I find this quite confusing.

Thank you for pointing this issue out. The illustration using the psychrometric chart is based on work by Monteith (1981). This diagram describes the magnitude of turbulent flux (length of arrow) at a view point from a parcel of air located at a reference height. Since the parcel of air receives heat and water vapour from the land surface, the final state is represented by the surface condition while the initial state is represented by the atmospheric conditions at the reference height. It should be noted that this diagram does not indicate a partial derivative of atmospheric state with respect to time (i.e., $\frac{\partial}{\partial t}$). Rather, the difference between the initial and the final states should be understood as the magnitude of the turbulent heat fluxes instead of changes in atmospheric state. We will explicitly address this point in the revised manuscript.

**16. Minor comment 11**
L120: "In the equilibrating process" - which process do you mean?

The equilibrating process indicates the second term in equations (2) and (3). We will explicitly describe the process in the next version.

**17. Minor comment 12**
L120: "the air parcel is adiabatically cooled" - why is it cooled/lifted? Isn't it rather by the mixing of the moistened air from the surface with the unsaturated air of the atmosphere due to buoyancy that depletes the difference?

We did not mention "lift" in the manuscript, and this concept is different from the "adiabatic lifting of air". Here, the adiabatic process indicates that there is no incoming energy into the system ($Q=LE+H=0$) (Monteith, 1981). We agree that this process is generated by the mixing of the air from the surface to the air to some level above the surface. The mixing driven by buoyancy is implicit in $r_a$ as we mentioned above (2. Major Comment1). We will try to revise this phrasing to reduce confusion.

**18. Minor comment 13**
L149: "rh budget" - relative humidity is not a conserved quantity, so there is no budgeting. If you talk about mass conservation, you would need to formulate this in terms of specific humidity or similar.

Thank you for pointing out the erroneous term. We will revise it in the next version. ("rh budget" → "rh changes with time")

**19. Minor comment 14**
L151: "This is logical in that LEG itself operates to diminish the vertical rh gradient." No, it does not. Vertical mixing is related to buoyancy, not to VPD. Although mixing also depletes the rh difference, it is not the same process.

We believe that the statement is not problematic. Please see our response above (2. Major Comment1).

**20. Minor comment 15**

L152/53: The classical equilibrium evaporation rate $LE = \frac{s}{s+\gamma} Q$ is not derived from the assumption of a saturated atmosphere. In the original derivation by Schmidt (1915), the only assumption is that the air immediately in contact with an open water surface is in thermodynamic equilibrium, so that the addition of energy is partitioned accordingly. But it does not assume that the atmosphere is saturated

Thank you for pointing out the original derivation by Schmidt (1915). We will refer the original derivation in the revised version of the manuscript. The thermodynamic equilibrium (or chemical equilibrium) between open water surface and the air in contact with the surface is equivalent to the saturation of the air (i.e., $rh_s = 1$). If the air at a reference height is not saturated under this condition, the second term of the Penman equation is required (equations (4) and (5)), and thus $\frac{s}{s+\gamma} Q$ alone cannot represent actual evaporation. de Bruin et al. (2016) also introduced the correction term $\beta$ due to unsaturated air (see 12. Minor comment 7). Therefore, our interpretation in lines 152/3 is consistent with the derivations we present.

**21. Minor comment 16**

L194-201: I don't understand the need for the wavelet analysis. Why is it necessary? It seems to me that it makes the analysis more complicated than necessary.

We believe the wavelet results provide useful information that support our interpretations. The primary purpose of this research is decomposing $LE$ into $LE_Q$ and $LE_G$ terms and identifying spatiotemporal variabilities of the two terms which yield behaviour of $LE$. The wavelet result shows how the variance of $LE$ is explained by $LE_Q$ and $LE_G$ terms in different time scales over specific period. For instance, the strong positive correlation between $LE$ and $LE_G$ in the longer time period in Figure 2 (d) demonstrates that $LE_G$ variability plays a non-negligible role in seasonal and interannual behaviour of $LE$. This is an unexpected result since the theory presented by McColl et al. (2019) implies zero seasonal variability of $LE_G$. We will try to better justify the use of wavelet and better interpret the results.

**22. Minor comment 17**

L203-212: I am skeptical about the use of daily averages. Relative humidity, wind speeds, air and surface temperatures, and aerodynamic conductance show pronounced variations at the diurnal scale. How do you account for the covariations among these variables if you use daily means?

We agree that sub-daily scale variabilities are important in this analysis. This is the reason why we highlighted Figure 3. Nevertheless, applying our decomposition method to daily averaged variables still provides useful information in that it can reveal seasonal and interannual variability of $LE$. As mentioned in the above response (21. Minor comment 16), seasonal variability of $LE_G$ plays an important role in seasonal and interannual behaviour of $LE$ which should be investigated further in different regions in the world. Decomposing $LE$ into $LE_G$ and $LE_Q$ at daily time scale and identifying spatiotemporal variability is important to validate and extend the theory presented by McColl et al. (2019).

**23. Minor comment 18**

L220-226: Same question: How do you account for covariations among variables when you use daily mean forcing?

Please see our response above (22. Minor Comment 17).

**24. Minor comment 19**

L229-234: I do not understand what the wavelet analysis should tell me. Why do you not simply use autocorrelations?

Please see our response above (21. Minor Comment 16).

**25. Minor comment 20**

L240: That LEG is close to zero in the absence of irrigation in 2016 supports the interpretation mentioned above that the second term in the PM equation relates to an advection effect.

Thank you for your insightful comment. We agree that $LE_G$ term can be related to local advection of sensible heat. Please see our response above (4. Major comment 3). We will discuss this interpretation in the next version.

**26. Minor comment 21**

L245: Figure 2: I would appreciate a little more information of the site - like precipitation input, solar radiation etc. to provide more background about the site.

We will add more information on the site in the next version of the manuscript or supplementary material.

**27. Minor comment 22**

L256: Does the case shown in Figure 3c represent a case with irrigation? Then, I guess, LEG relates to advection effects?

Thank you for your comment. Figure 3(c) does not represent a case with irrigation (pre-harvest period). However, we still agree that a positive $LE_G$ term can be related to the local advection of sensible heat. Please see our response above (4. Major comment 3). We will discuss this interpretation in the next version.

**28. Minor comment 23**

L265: Figure 3: I would find it informative to also see H and net radiation, as well as the diurnal variations in the rh's.

We will add those variables in the next version of the manuscript or supplementary material.

**29. Minor comment 24 and 25**

L286: The statement that "the land surface is generally under thermodynamic equilibrium with the atmosphere at the global-annual scale" is, I think, incorrect. The finding that $rh_s \approx rh_a$ simply means that the air near the surface is well mixed, likely due to buoyancy.

L322: "land-atmosphere equilibrium is achieved ..." - again, the authors neglect the role of buoyancy that mixes the air near the surface here and which is likely to play the dominant role in reducing the relative humidity difference.

Please see our response above (2. Major Comment 1).

**30. Minor comment 26 and 27**

L329-334: I think the implications need to be rethought, given that the role of buoyancy in depleting a difference in rh has been neglected.

L335: Conclusions - same here, given the methodological flaws of the study, this paragraph needs to be rethought.

We will revise the discussion and conclusion section, and the role of buoyancy will be acknowledged. Please see also our response above to (2. Major Comment 1).

[revised manuscript text omitted]

---

## Author Response (AR1)

**Response to Reviewers' Comments**

In the following, we present our detailed responses to reviewers' suggestions with our replies in blue; the specific revisions that we performed are indicated by underlined text. In this response, we reconstructed comments by Reviewer #1 based on the last paragraph's suggestions.

**Reviewer #1**

**1. General comment**

This interesting paper explores a formulation of ET fully independent of the surface resistance, relying instead on a humidity resistance depending on the gradient of moisture between the surface and screen level, following the ideas displayed in Monteith (1981). Conceptually this approach allows to skip any explicit dependency of the characteristics of the surface, particularly those of the vegetation. The approach is worth exploring and this manuscript intends to show us to what point it can be used, especially for interpretation of the processes in place, more than in a parameterisation mode.

In a set of recent papers McColl and colleagues showed that, for scales of a day or longer, ET is essentially determined by the balance between surface moistening and surface heating, with the consequence at these temporal scales that more atmospheric moisture is the result of more ET, and the term describing it is equivalent to the diabatic term in the Penman-Monteith equation, therefore independent of the characteristics of the surface.

In this manuscript, the authors place themselves in this framework and try to extend it to sub-daily scale by using the humidity resistance in the adiabatic term, avoiding to prescribe any characteristic of the surface (vegetated or not). Their theoretical reasoning is easy to follow, substituting the pressure vapour by the relative humidity, and it is intended to be valid even for non-saturated surfaces through the prescription of a "surface relative humidity", $rh_s$.

We thank Reviewer #1 for your interest and constructive feedback on the manuscript. Your summary is consistent with what we wish to communicate regarding our study.

**2. Specific suggestion 1**

*Suggestions:* clarify somehow the aims of their research in the initial parts of the paper, especially in the abstract, when one may get the impression that a new ET parameterization is presented, while in reality what we have is a nice method of analysis of the diabatic and adiabatic components of ET;

*Critique:* The proposed method cannot be used as an independent way to determine $LE$, because $LE$ is used to determine $rh_s$, so obtaining again $LE$ from this $rh_s$ value would be of no use (unless I miss something). However the method is useful to separate the observed $LE$ into its diabatic and adiabatic parts (taking into account that $Q=H+LE$ is a tough assumption), and this is where most of the interpretation effort is put on.

Thank you for your suggestion. We agree with your suggestion. We focused on diagnostic analyses instead of predictive analyses using the proposed PM$_{rh}$ model in order to demonstrate the importance of partitioning $LE$ into diabatic and adiabatic components to improve understanding of the surface energy balance. Therefore, we have revised the abstract (L20-21) and the last paragraph of the introduction (L71-77) to explicitly highlight the decomposition analysis.

In this paper, we applied the new PM$_{rh}$ model for a diagnostic purpose (i.e., using the model to aid in interpretation of governing mechanisms of measured or predicted ET). As such, $rh_s$ was determined from

measured $LE$ and $H$ in this study. However, it should be noted that the proposed $PM_{rh}$ model has the potential to be used to determine ET in principle if a model to determine $rh_s$ is available, although this approach is beyond the scope of the present study. For instance, a combination of surface temperature and soil moisture (remotely sensed or field measurement) can be used to predict $rh_s$. We have described this point in the discussion section to suggest potential applications of the proposed model in the future (section 5.3: L412-430).

**3. Specific suggestion 2**
*Suggestions:* elaborate on the meaning of $rh_s$;

*Critique:* The authors do not explicitly comment or define $rh_s$, which is still the missing piece in most of the approaches dealing with non-saturated surfaces. In the present case, they go around the problem of defining or calculating $rh_s$ by deriving a value for it by using observed $LE$ and $H$ values from EC-systems, using their formulation (2).

$rh_s$ is the relative humidity at the land surface, and is a purely physical quantity (vapour pressure divided by saturation vapour pressure). Here, the land surface can be defined as a single plane located at $d+z_{0h}$ (d=displacement height, $z_{0h}$ = roughness length for heat) following the bigleaf framework of micrometeorology (Knauer et al., 2018a). We noticed that the land surface was not explicitly defined in the preprint version, and thus we have added the definition of $rh_s$ and the land surface explicitly in the theory section by introducing a new subsection (section 2.1: L84-90 describes the meaning of $rh_s$).

**4. Specific suggestion 3**
*Suggestions:* explain better what are the expected consequences of their hypotheses in the ulterior data analysis (such as imposing $Q=H+LE$ or the chosen form for $r_a$);

*Critique:* To do it they assume that the available energy $Q$ is $LE+H$ instead of $R_n$-$G$ trying to circumvent the unavoidable problem of the closure of the surface energy budget. This decision could be understandable but it is poorly justified and the consequences of it are not reflected upon. Unfortunately the result of these strong hypotheses concerning $rh_s$ and a discussion of the values obtained is not explicitly shown or made.

Thank you for pointing out this problem. We agree that the well-known surface energy balance closure problem for eddy covariance (EC) observations ($R_n - G > H+LE$) can cause systematic uncertainty in our analysis. We determined that most of the energy imbalance in our site is contributed by unmeasured canopy and soil heat storages. Although we cannot exactly quantify these storage terms, we reason that this is the primary source of energy imbalance as follows: First, it is expected that unmeasured canopy and soil heat storages in this site are significant since the sugarcane canopy grows up to 3.6 m tall with a dense canopy. Indeed, when canopy height was less than 1 m, the surface energy balance was very nearly closed (97%), whereas the closure was 83 % when canopy height was higher than 1 m. This result supports our reasoning concerning canopy storage terms. Also, it is widely accepted that the influence of secondary circulations on the energy balance closure is small for a homogenous landscape (Mauder et al., 2020;Stoy et al., 2013;Leuning et al., 2012). Since our site is located within a homogenous landscape, the influence of secondary circulations on the energy balance closure may be negligible. Even if the lack of energy balance is due to underestimation of $LE + H$, there is no consensus on a universally appropriate method to correct $LE$ and $H$ (Mauder et al., 2020). Therefore, we did not force energy balance closure for the Costa Rica site. These points have been more explicitly described in the updated manuscript in L208-219, and a satellite picture that shows the homogenous landscape is provided in Figure S1.

Wehr and Saleska (2021) recently demonstrated that regardless of whether the lack of energy balance of EC observation is due to $LE + H$ or due to $R_n - G$, applying the flux gradient equation to the observed $LE$

and $H$ without applying an energy balance correction is the best approach to determining surface resistance (conductance). This is because applying the flux gradient equation to the observed $LE$ and $H$ dispenses with the unnecessary assumption of energy balance closure (i.e., $LE + H = R_n – G$). They showed that bias introduced by underestimated $LE$ and $H$ is smaller than the bias introduced by the energy balance closure assumption. This finding may be applied to our analysis, although in our case we are calculating $rh_s$ instead of surface conductance. This is another reason why we imposed $Q$ as $H+LE$ and do not force energy balance closure.

As for the FLUXNET dataset, we provided an analysis using energy balance corrected $LE$ and $H$ (Bowen ratio preserving method in Pastorello et al. (2020)) in the supplementary file; the results for corrected and uncorrected versions were almost identical. This is actually a natural consequence. In equation (10) of the revised version, $LE$ and $H$ are included in the numerator and denominator respectively. Multiplying the same ratio to $LE$ and $H$ in equation (10) to correct $LE$ and $H$ based on the Bowen ratio method does not significantly change the resulting $rh_s$. Therefore, the lack of surface energy balance closure does not significantly impact our analyses and interpretations unless the lack of energy balance is dominated by $LE$ only or by $H$ only. If the lack of energy balance is dominated by $LE$ only or by $H$ only, our results and interpretation could include systematic bias, representing a shortcoming of our approach.

As for the chosen form of $r_a$, the influence of this choice is also expected to be marginal compared to the energy balance problem. Knauer et al. (2018b) showed that uncertainty caused by different $r_a$ on surface conductance is low compared to the energy balance closure problem. This finding can be applied to our analysis. Specifically, in equation (10), $r_a$ is multiplied by both denominator and numerator, and thus a small difference in $r_a$ should not significantly affect the resulting $rh_s$.

We have added discussion points regarding the energy balance closure problem and chosen form for aerodynamic resistance in L441-463 in the revised version.

**5. Specific suggestion 4**
*Suggestions:* expand they interpretation of data, currently very shallow, into a remade Discussion section;

*Critique:* Fig 4 contains a lot of information in its 8 sub-figures, which are not really commented. A similar comment can be made about Fig 5 and its 12 sub-figures. What is the use of displaying so much information if then it is not discussed? In general section 4 would need to be more developed in terms of interpretation of results, which is now very shallow, especially sections 4.2 and 4.3.

Thank you for your suggestion. In the revised version of the manuscript, we have expanded our interpretation in the result section, especially section 4.2 and 4.3 (L309-358). We have tried to discuss all sub-figures in a consistent way. Also, we have remade the Discussion section which includes four sub section highlighting sub-daily scale interpretation, longer time scale interpretation, potential applications, and limitations (L372-463).

**6. Specific suggestion 5**
*Suggestions:* joint the current "Discussion" and "Conclusions" sections into a more comprehensive and developed new "Conclusions" section;

*Critique:* As the paper is now sections 5 "Discussion" and 6 "Conclusions", both very short, could be merged into one larger Conclusions section. Instead a real "Discussion" section could come from an expanded version of the analysis of the results.

Thank you for your suggestion. We have remade the Discussion section as mentioned above and revised the Conclusions (L465-476).

**7. Specific suggestion 6**
*Suggestions:* consider to summarise the information in the supplementary material and incorporate it straight into the manuscript

We felt that providing detailed information on the field observations is valuable for this paper, and thus we prepared the supplementary file. The reason we structured the presentation as the main paper plus a supplementary file is that the manuscript already covers a lot of information (e.g., new theory, and results from three different datasets), and incorporating supplementary material directly into the manuscript may overwhelm some readers. Instead of incorporating the supplementary material, we remove the original Appendix and incorporate the information into the theory section of the main paper in order to reduce the complicated structure of the paper (L79-111).

**Reviewer #2**

**1. General comment**

Recommendation: major revision or reject with encouragement to resubmit

This manuscript deals with the role of relative humidity in evaporation estimates. It presents a novel approach to evaluate this role, and as such, I think this is very valuable. However, there are a couple of issues, some of which are major, that need to be addressed. I also had a hard time to follow the paper, so I think the authors need to work on a better structure. Also, the authors do not discuss any shortcomings of their study, showing a lack of critically assessing their own findings. There are certainly quite a few, as shown below, and these need to be assessed and discussed before any conclusions can be drawn from the analysis. As some of the major issues described below are likely to substantially change this manuscript in order to be addressed, at least major revisions need to be made, or, alternatively, the manuscript could be rejected with encouragement to resubmit, so that it would again enter an open review discussion.

We thank Reviewer #2 for the constructive feedback on the manuscript. We have endeavored to improve the structure of the manuscript and address the issues highlighted by the Reviewer in a revised manuscript. Below we provide point by point responses to issues raised along with how we will improve our manuscript based on your suggestions.

**2. Major comment 1**

"Land-atmosphere equilibrium". The authors describe that the relative humidity difference between the surface and the atmosphere is a key driver for evaporation, and its depletion is an indication of thermodynamic equilibrium. This picture is incorrect. The water vapor transport from the surface into the atmosphere is driven primarily by buoyancy, that is, by the sensible heat flux, not by the difference in vapor pressure. Hence, a depletion of the relative humidity difference is solely a reflection of well-mixed air near the surface. This misconception is reflected in much of the manuscript, with buoyant mixing not mentioned anywhere, so this needs to be addressed in a revision.

Thank you for pointing out the issue. In this criticism, the Reviewer implies that "equilibrium" is a concept related to atmospheric stability or generating vertical motion of a fluid. From this point of view, our "land-atmosphere equilibrium" would indeed be a misconception since it is not related to generating turbulent mixing. However, this interpretation overlooks other types of thermodynamic equilibria. Although the "equilibrium" term in this manuscript is not related to generating turbulence, it is tightly related to chemical equilibrium (Kleidon et al., 2009) as well as steady-state equilibrium evaporation inside an atmospheric boundary layer (ABL) model (McColl et al., 2019).

We agree that a depletion of the *rh* difference reflects well-mixed air near the surface. Nevertheless, this well-mixed situation can be understood as an "equilibrium" condition without representing a misconception. From a thermodynamic point of view, the $LE_G$ term associated to the dissipation of the *rh* vertical gradient can be seen as a process producing entropy by bringing the chemical potential of the surface water to that of the atmosphere (Kleidon et al., 2009). Also, the term "equilibrium" in boundary layer meteorology is dynamically defined as the steady-state of an ABL (McNaughton and Jarvis, 1983;Raupach, 2001;McColl et al., 2019). For instance, McColl et al. (2019) stated that *"While there are various definitions of equilibrium ET, arguably the most fundamental is the evaporative state achieved by a closed system forced with constant incoming radiation that is partitioned at the lower boundary between latent and sensible heat fluxes"*. As we addressed in L172~191 in the updated manuscript, evaporation approaches a steady-state of equation (8) in an ABL model (McColl et al., 2019). We showed that this condition is equivalent to a depletion of the *rh* difference. Therefore, the proposed PM$_{rh}$ model can be understood as an extension of the new theory by McColl et al. (2019); we highlight this point in relation to the equilibrium concept as presented in our study.

We agree that buoyancy generated by temperature gradients enhances vertical motion on top of the mechanic turbulence increasing diffusivity of eddies transporting both heat and vapour. To account for diffusivity in the flux gradient equation, buoyancy driven vertical motion is parameterized in aerodynamic resistance ($r_a$), which includes a thermal stability correction function (see equation (9) of the revised version). Our derivation of PM$_{rh}$ model stems from the flux gradient equation, and thus buoyancy-driven energy is implicit in $r_a$. It should be noted that the $rh$ difference between the land and the atmosphere does not affect $r_a$ in our framework. In the revised manuscript, we more explicitly described $r_a$ along with a mention of buoyancy considerations to reduce confusion by incorporating derivation of the PM$_{rh}$ model from Appendix to the theory section (L79-111).

**3. Major comment 2**

Shortcomings of the PM equation. The authors use the Penman Monteith (PM) equation as the basis of their work, although this equation has some clear deficiencies. This was, for instance, clearly shown in the paper by Milly and Dunne (2016, "Potential evapotranspiration and continental drying", Nature Climate Change), also by Renner et al. (2019, "Using phase lags to evaluate model biases in simulating the diurnal cycle of evapotranspiration: a case study in Luxembourg", HESS). The authors take the PM equation for granted, but I think they need to be more critical and evaluate potential flaws in their work in light of these shortcomings.

We agree that the PM equation has several limitations. However, we did not intend to take the PM equation for granted. Rather, the proposed PM$_{rh}$ model is introduced in order not to include surface resistance in the evaporation model; surface resistance is the key parameter of the PM equation (please see L64~70). The above-referred studies (Milly and Dunne, 2016;Renner et al., 2019) showed that bias introduced by the PM equation mostly originates from inaccurate representation of surface resistance (or conductance), and our proposed PM$_{rh}$ model does not include surface resistance as a means to overcome this shortcoming in the PM equation.

Of course, our proposed PM$_{rh}$ model shares other shortcomings with the PM equation, such as the linearization of the saturation vapour pressure slope, the assumption of identical aerodynamic resistances for sensible heat and water vapour, and the assumption that the land surface can be treated as a single plane (i.e., bigleaf). We agree that these shortcomings should be acknowledged in the manuscript. We explicitly discuss these shortcomings of the PM equation in the updated version as limitations that also apply to the proposed PM$_{rh}$ model (L432-440).

**4. Major comment 3**

Role of advection. Literature by de Bruin (e.g., de Bruin et al., 2016, "A Thermodynamically Based Model for Actual Evapotranspiration of an Extensive Grass Field Close to FAO Reference, Suitable for Remote Sensing Application", J. Hydromet.) shows that it is basically the equilibrium evaporation term that dominates evaporation rates, and that the second term comes from advection, e.g. in the case of evaporation from irrigated land in an arid region. Given that the evaluation includes data from Costa Rica from an irrigated site during the dry season, advection could play quite an important role in shaping the evaporation estimate. But as far as I can tell, advection is not being mentioned as a phenomenon in the manuscript. In addition to the shortcomings of the PM equation, I think there is a good reason to doubt the analysis so these factors need to be assessed.

Thank you for pointing out this important issue. Under local advection of sensible heat (i.e., horizontal sensible heat advection from an adjacent dry field), eddy covariance (EC) observations no longer represent net fluxes from a control volume (Leuning, 2004). In these conditions, the energy balance equation ($LE + H = R_n - G$) for the control volume may be rewritten as $LE + H = R_n - G + Q_{adv}$ (de Bruin et al., 2016), where $Q_{adv}$ represents the sensible heat horizontally advected.

In our sugarcane site in Costa Rica, however, there is no evidence that local advection plays an important role. The flux observation plot was surrounded by similar sugarcane agriculture which also utilizes irrigation in dry season. Around this homogenous landscape, effects of local advection can be marginal (Leuning et al., 2012). We added a satellite image of the surrounding landscape in the revised version of the supplementary material (Figure S1). Also, daily mean $H$ was rarely negative in dry season regardless of irrigation application. Negative values of $H$ are used as an indicator of local advection (Kutikoff et al., 2019). Therefore, local advection of sensible heat may not be significant in our study site. We have added daily $H$ time series in the Figure 2 in the revised version.

Further, even if local advection plays a non-negligible role, our decomposition approach (i.e., $LE$ into $LE_Q$ and $LE_G$) is not affected by $Q_{adv}$ in principle. Since we defined available energy ($Q$) as $LE + H$ instead of $R_n - G$ in calculating $rh_s$, the decomposition approach is simply determined by $LE$ and $H$ using the EC technique, regardless of whether $LE + H = R_n - G + Q_{adv}$ or $LE + H = R_n - G$. Therefore, even under local advection conditions, the decomposition approach is robust. However, one of the assumptions of the $PM_{rh}$ model, identical eddy diffusivities for water vapour and sensible heat, can be problematic under strong local advection conditions (Lee et al., 2004). This limitation has been discussed as a shortcoming of our approach in the revised version (L435-438).

If $Q_{adv}$ plays a non-negligible role, it should increase $LE_G$ term in principle as the Reviewer also argued in the minor comments (see 12. Minor comment 7, 25. Minor comment 20, 27. Minor comment 22). This is because local advection of sensible heat is typically accompanied by negative $H$ (de Bruin and Trigo, 2019), which implies a large $LE_G$ value. We agree with these points and the influence of advection on $LE_G$ term has been discussed in the revised manuscript (L387-394).

**5. Major comment 4**

Closure of the energy balance. The authors write on L177 that they did not enforce an energy balance closure, attributing the lack of closure due to unmeasured heat storage terms (but without quantifying and supporting this attribution). However, I do not think that this is a feasible way to do this analysis. It seems to me that the scientific consensus is that the imbalance is mostly attributable to secondary circulations in the convective boundary layer (see, e.g., the review by Mauder et al., 2020, Boundary Layer Meteorology, https://doi.org/10.1007/s10546-020-00529-6 ). I think this aspect also needs to be addressed thoroughly in the analysis.

Thank you for pointing out this problem. We agree that the well-known surface energy balance closure problem for eddy covariance (EC) observations ($R_n - G > H+LE$) can cause systematic uncertainty in our analysis. We determined that most of the energy imbalance in our site is contributed by unmeasured canopy and soil heat storages. Although we cannot exactly quantify these storage terms, we reason that this is the primary source of energy imbalance is as follows: First, it is expected that unmeasured canopy and soil heat storages in this site are significant since the sugarcane canopy grows up to 3.6 m tall with a dense canopy. Indeed, when canopy height was less than 1 m, the surface energy balance was very nearly closed (97%), whereas the closure was 83 % when canopy height was higher than 1 m. This result supports our reasoning concerning canopy storage terms. Also, it is widely accepted that the influence of secondary circulations on the energy balance closure is small for a homogenous landscape (Mauder et al., 2020;Stoy et al., 2013;Leuning et al., 2012). Since our site is located within a homogenous landscape, the influence of secondary circulations on the energy balance closure may be negligible. Even if the lack of energy balance is due to underestimation of $LE + H$, there is no consensus on a universally appropriate method to correct $LE$ and $H$ (Mauder et al., 2020). Therefore, we did not force energy balance closure for the Costa Rica site. These points have been more explicitly described in the updated manuscript in L208-219, and a satellite picture revealing a homogenous landscape was attached in Figure S1.

Wehr and Saleska (2021) recently demonstrated that regardless of whether the lack of energy balance of EC observation is due to $LE + H$ or due to $R_n – G$, applying the flux gradient equation to the observed $LE$ and $H$ without applying an energy balance correction is the best approach to determining surface resistance (conductance). This is because applying the flux gradient equation to the observed $LE$ and $H$ dispenses with the unnecessary assumption of energy balance closure (i.e., $LE + H = R_n – G$). They showed that bias introduced by underestimated $LE$ and $H$ is smaller than the bias introduced by the energy balance closure assumption. This finding may be applied to our analysis, although in our case we are calculating $rh_s$ instead of surface conductance. This is another reason why we imposed $Q$ as $H+LE$ and do not force energy balance closure.

As for the FLUXNET dataset, we provided an analysis using energy balance corrected $LE$ and $H$ (Bowen ratio preserving method in Pastorello et al. (2020)) in the supplementary file; the results for corrected and uncorrected versions were almost identical. This is actually a natural consequence. In equation (10) of the revised version, $LE$ and $H$ are included in the numerator and denominator respectively. Multiplying the same ratio to $LE$ and $H$ in equation (10) to correct $LE$ and $H$ based on the Bowen ratio method does not significantly change the resulting $rh_s$. Therefore, the lack of surface energy balance closure does not significantly impact our analyses and interpretations unless the lack of energy balance is dominated by $LE$ only or by $H$ only. If the lack of energy balance is dominated by $LE$ only or by $H$ only, our results and interpretation could include systematic bias, representing a shortcoming of our approach.

We have added discussion points regarding the energy balance closure problem in L441-459 in the revised version.

**6. Minor comment 1**
Title: "relative humidity gradients" are not a constraint. They are highly dependent on moisture and heating of air, hence not an independent variable, and they are not a physical constraint, such as those imposed by the energy- or water balance, or the laws of thermodynamics.

We agree that relative humidity is a physical quantity that depends on both moisture and heat of air. However, the main finding of this study and implication of the proposed model is that vertical relative humidity gradients constrain the latent heat flux (and thus evapotranspiration). This has not been recognized previously, but we feel that we have demonstrated this point in the present study.

**7. Minor comment 2**
L34: "LE predictions remain highly uncertain" - I doubt this statement. The basic constraints for evaporation have been quite well established over decades, so what are the factors that remain uncertain? The authors need to be more specific than this wide-sweeping claim.

Thank you for your suggestion. However, this sentence has been removed in the revised version of the manuscript.

**8. Minor comment 3**
L38: Actually, the "pioneering work" on governing physics of LE started with Schmidt (1915), as described by de Bruin et al (reference provided above).

Thank you for your suggestion. We have acknowledged Schmidt (1915) in the revised version (L36).

**9. Minor comment 4**
L46: "rh budget" - there is no rh budget, because relative humidity is not a conserved quantity, as it jointly depends on temperature and moisture content. So you can talk about an energy budget or a moisture budget, but not of a rh budget.

Thank you for pointing out the erroneous term. We have revised the sentence as "They hypothesized that changes in relative humidity (*rh*) with respect to time is roughly steady in an idealized atmospheric boundary layer at daily to monthly timescale." (L45-47)

**10. Minor comment 5**
L51: I would not describe the PM equation as reflecting governing physics, it is at best semi-empirical, with shortcomings (see above, major comment 2).

We agree that PM equation is semi-empirical although some of its physics is correct. We have revised this sentence. ("understand the governing physics of" → "express") (L51)

**11. Minor comment 6**
L69-83: I do not understand the derivation (and it seems odd to have this derivation partly in the introduction and partly in the appendix). What does surface relative humidity stand for? When we deal with a vegetated surface that experiences water limitation, then any water that does evaporate either diffuses out of the soil (for which I would think a surface resistance would be rather critical to consider) or it is evaporated inside a leaf, but with the exchange with the atmosphere constrained by stomatal conductance. So how do these equations (2) and (3) reflect a physical picture of the evaporation process?

In order to clarify the $PM_{rh}$ model and its derivation, we have made a new section in the revised version of the manuscript (section 2.1: L79-111).

The surface relative humidity ($rh_s$) represents a physical quantity, relative humidity (i.e., vapour pressure divided by saturation vapour pressure) at the land surface. Here, the land surface is defined as a single plane located at $d+z_{0h}$ (d=displacement height, $z_{0h}$ = roughness length for heat) following the bigleaf framework of micrometeorology (Knauer et al., 2018a). We added the definitions of the relative humidity and the land surface explicitly in L84-90.

The proposed $PM_{rh}$ model (equations (2) and (3)) conceptually bypasses the need to characterize surface resistance in order to describe water vapour transport and it is the key novelty of the model. The $PM_{rh}$ model is not impacted by whether the water vapour flux originates from the soil or from inside stomata of vegetation. Instead, the model only describes exchange of water vapour and heat from the land surface to the atmosphere through a turbulent process, which is parameterized by $r_a$. The focus of the model is to decompose water vapour exchange into two thermodynamic processes following the notion of Monteith (1981). In this way, evaporation can be understood as a combination of diabatic and adiabatic processes.

**12. Minor comment 7**
Eq (4): As pointed out in the major comments, the second term in the PM equation is likely reflecting advective conditions (see paper by de Bruin et al., mentioned earlier).

We agree that a large resulting value for the second term in the Penman equation can be an indication of advection. However, it is worth noting that even under the advection-free conditions, the second PM term is still required. de Bruin et al. (2016) expressed wet surface (irrigated grass) evaporation under advection-free conditions as $LE = \frac{s}{s+\gamma} Q + \beta$, where, $s$ is the linearized slope of saturation vapour pressure versus temperature, $\gamma$ is the psychrometric constant, $Q$ is the available energy, and $\beta$ is a constant correction. They introduced $\beta$ due to unsaturated air inside the atmospheric boundary layer even under advection-free conditions due to entrainment. $\beta$ should be understood as the second term of the Penman equation in this case. We have discussed the role of local advection of sensible heat and the entrainment in L387-394.

**13. Minor comment 8**

L103: "When the vertical gradient of rh dissipates...": see Major Comment 1.

Please see our response above (2. Major Comment1); we do not believe this expression to be a misconception.

**14. Minor comment 9**

L114: Again, I do not understand the reasoning of the authors (see comment above, L69-83).

Please see our response above (11. Minor comment 6); the focus of the proposed $PM_{rh}$ model is to decompose water vapour exchange into two thermodynamic processes following the notion of Monteith (1981). In this section of our manuscript, we interpreted the $PM_{rh}$ model using the psychrometric chart. Interpretation of evaporation as two thermodynamic processes using the psychrometric chart is a widely accepted approach (Monteith, 1981;Monteith and Unsworth, 2013;Monson and Baldocchi, 2014). In order to clarify our reasoning, we have described the background of the concept more explicitly in the revised version (L139-146).

**15. Minor comment 10**

Figure 1: Why is the initial state represented by the atmospheric conditions, and the final state represented by the surface conditions? Doesn't evaporation start at the surface? Actually, evaporation may already start within the soil (bare soil evaporation) or inside leaves. How does this fit into this diagram? I find this quite confusing.

Thank you for pointing this issue out. The illustration using the psychrometric chart is based on the original work by Monteith (1981). This diagram describes the magnitude of turbulent flux (length of arrow) at a view point from a parcel of air located at a reference height. Since the parcel of air receives heat and water vapour from the land surface, the final state is represented by the surface condition while the initial state is represented by the atmospheric conditions at the reference height. It should be noted that this diagram does not indicate a partial derivative of atmospheric state with respect to time (i.e., $\frac{\partial}{\partial t}$). Rather, the difference between the initial and the final states should be understood as the magnitude of the turbulent heat fluxes instead of changes in atmospheric state. We have explicitly addressed this point in the revised manuscript (L139-146) and the caption of Figure 1.

**16. Minor comment 11**

L120: "In the equilibrating process" - which process do you mean?

The equilibrating process indicates the second term of the $PM_{rh}$ equation. We have revised the expression as "In the *rh* equilibrating process" to clarify the concept (L148).

**17. Minor comment 12**

L120: "the air parcel is adiabatically cooled" - why is it cooled/lifted? Isn't it rather by the mixing of the moistened air from the surface with the unsaturated air of the atmosphere due to buoyancy that depletes the difference?

We did not mention "lift" in the manuscript, and this concept is different from the "adiabatic lifting of air". Here, the adiabatic process indicates that there is no incoming energy into the system ($Q=LE+H=0$) (Monteith, 1981). We agree that this process is generated by the mixing of the air from the surface to the air to some level above the surface. The mixing driven by buoyancy is implicit in $r_a$ as we mentioned above (2. Major Comment1). In order to clarify the concept, we have revised the phrasing as "the air parcel is adiabatically cooled (or heated when $rh_s < rh_a$) due to turbulent mixing" (L148).

**18. Minor comment 13**

L149: "rh budget" - relative humidity is not a conserved quantity, so there is no budgeting. If you talk about mass conservation, you would need to formulate this in terms of specific humidity or similar.

Thank you for pointing out the erroneous term. We revised the sentence as "They hypothesized that in many continental regions, the near surface atmosphere is in state of equilibrium, where $rh$ is steady with time in an idealized atmospheric boundary layer at longer than daily time scales." (L178-180)

**19. Minor comment 14**

L151: "This is logical in that LEG itself operates to diminish the vertical rh gradient." No, it does not. Vertical mixing is related to buoyancy, not to VPD. Although mixing also depletes the rh difference, it is not the same process.

We do not believe that the statement is problematic. Please see our response above (2. Major Comment1). To reduce confusion, we slightly revised the sentence as "This is logical in that $LE_G$ itself diminish the vertical $rh$ gradient over the course of a day." (L182-183)

**20. Minor comment 15**

L152/53: The classical equilibrium evaporation rate $LE = \frac{s}{s+\gamma} Q$ is not derived from the assumption of a saturated atmosphere. In the original derivation by Schmidt (1915), the only assumption is that the air immediately in contact with an open water surface is in thermodynamic equilibrium, so that the addition of energy is partitioned accordingly. But it does not assume that the atmosphere is saturated

Thank you for pointing out the original derivation by Schmidt (1915). We have referred the original derivation in the revised version of the manuscript. The thermodynamic equilibrium (or chemical equilibrium) between open water surface and the air in contact with the surface is equivalent to the saturation of the air (i.e., $rh_s = 1$). If the air at a reference height is not saturated under this condition, the second term of the Penman equation is required (equations (4) and (5)), and thus $\frac{s}{s+\gamma} Q$ alone cannot represent actual evaporation. Therefore, some previous authors define the classical equilibrium evaporation as evaporation from a saturated surface into saturated air. Also, there are several definitions for the classical equilibrium evaporation (Raupach, 2001), and the above definition is one of them. To clarify this point, we have revised the sentence as "This is consistent with one of the classical definitions of equilibrium $LE$ that defines equilibrium $LE$ as evaporation from a saturated surface into saturated air" (L189-190).

**21. Minor comment 16**

L194-201: I don't understand the need for the wavelet analysis. Why is it necessary? It seems to me that it makes the analysis more complicated than necessary.

We believe the wavelet results provide useful information that support our interpretations. The primary purpose of this research is decomposing $LE$ into $LE_Q$ and $LE_G$ terms and identifying spatiotemporal variabilities of the two terms which yield behaviour of $LE$. The wavelet result shows how the variance of $LE$ is explained by $LE_Q$ and $LE_G$ terms in different time scales over specific period. For instance, the strong positive correlation between $LE$ and $LE_G$ in the longer time period in Figure 2 (d) demonstrates that $LE_G$ variability plays a non-negligible role in seasonal and interannual behaviour of $LE$. This is an unexpected result since the theory presented by McColl et al. (2019) implies zero seasonal variability of $LE_G$.

**22. Minor comment 17**

L203-212: I am skeptical about the use of daily averages. Relative humidity, wind speeds, air and surface temperatures, and aerodynamic conductance show pronounced variations at the diurnal scale. How do you account for the covariations among these variables if you use daily means?

We agree that sub-daily scale variabilities are important in this analysis. This is the reason why we highlighted Figure 3. Nevertheless, applying our decomposition method to daily averaged variables still provides useful information in that it can reveal seasonal and interannual variability of $LE$. As mentioned in the above response (21. Minor comment 16), seasonal variability of $LE_G$ plays an important role in seasonal and interannual behaviour of $LE$ which should be investigated further in different regions in the world. Decomposing $LE$ into $LE_G$ and $LE_Q$ at daily time scale and identifying spatiotemporal variability is important to validate and extend the theory presented by McColl et al. (2019). Also, $LE_G$ at daily time scale can be understood as an indicator reflecting land surface dryness relative to the atmosphere. We have highlighted this point in the revised version of the manuscript (L184-187).

**23. Minor comment 18**
L220-226: Same question: How do you account for covariations among variables when you use daily mean forcing?

Please see our response above (22. Minor Comment 17).

**24. Minor comment 19**
L229-234: I do not understand what the wavelet analysis should tell me. Why do you not simply use autocorrelations?

Please see our response above (21. Minor Comment 16). We tried to extend our interpretation on the wavelet analysis in the revised version (L275-279).

**25. Minor comment 20**
L240: That LEG is close to zero in the absence of irrigation in 2016 supports the interpretation mentioned above that the second term in the PM equation relates to an advection effect.

Thank you for your insightful comment. We agree that $LE_G$ term can be related to local advection of sensible heat. Please see our response above (4. Major comment 3). We have discussed this interpretation in the revised version (L387-394).

**26. Minor comment 21**
L245: Figure 2: I would appreciate a little more information of the site - like precipitation input, solar radiation etc. to provide more background about the site.

We have added monthly precipitation and daily heat flux in Figure 2 in the revised version.

**27. Minor comment 22**
L256: Does the case shown in Figure 3c represent a case with irrigation? Then, I guess, LEG relates to advection effects?

Thank you for your comment. Please note that Figure 3(c) does not represent a case with irrigation (pre-harvest period). However, we still agree that a positive $LE_G$ term can be related to the local advection of sensible heat. Please see our response above (4. Major comment 3).

**28. Minor comment 23**
L265: Figure 3: I would find it informative to also see H and net radiation, as well as the diurnal variations in the rh's.

We have added diurnal cycle of $H$ and $rh_s$ in Figure 3.

**29. Minor comment 24 and 25**
L286: The statement that "the land surface is generally under thermodynamic equilibrium with the atmosphere at the global-annual scale" is, I think, incorrect. The finding that $rh_s \approx rh_a$ simply means that the air near the surface is well mixed, likely due to buoyancy.

L322: "land-atmosphere equilibrium is achieved ..." - again, the authors neglect the role of buoyancy that mixes the air near the surface here and which is likely to play the dominant role in reducing the relative humidity difference.

Please see our response above (2. Major Comment 1). We have tried to clarify the implications of the findings by remaking the discussion section.

**30. Minor comment 26 and 27**
L329-334: I think the implications need to be rethought, given that the role of buoyancy in depleting a difference in rh has been neglected.

L335: Conclusions - same here, given the methodological flaws of the study, this paragraph needs to be rethought.

We have revised the discussion and conclusion section, and the role of buoyancy was acknowledged in the theory section. Please see also our response above to (2. Major Comment 1).

---

## Author Response (AR2)

**Response to Reviewers' Comments (second round)**

In the following, we present our detailed responses to reviewers' suggestions with our replies in blue; the specific revisions that we performed are indicated by underlined text.

**Reviewer #1**

Dear authors

Thanks for taking in consideration most of my comments. I still find the issue of the surface relative humidity (rhs) treated in a very shallow manner. Please consult the recent paper https://doi.org/10.1007/s10546-020-00550-9, where a review and a discussion on the meaning and computation of rhs is included.

We thank Reviewer #1 for constructive feedback on the manuscript. Following your suggestion, we have revised the paragraph as follow (L88-92).

"... where  $rh_s$  is surface relative humidity, i.e., the ratio of  $e_s$  to  $e^*(T_s)$ . For a vegetated surface,  $rh_s$  as defined in this study represents relative humidity of the foliage surface and is conceptually equivalent to surface water availability in Li and Wang (2019). For a bare soil land surface,  $rh_s$  represents soil surface relative humidity which can be found using the "alpha" method that is parameterized using soil moisture content or soil water potential (Lee and Pielke, 1992; Wu et al., 2000; Cuxart and Boone, 2020). "

**Reviewer #2**

I have read the authors' response as well as the revision to the manuscript. Even though I do not agree with some of the interpretations, I think that this version is suitable for acceptance after minor revisions.

We thank Reviewer #2 for the constructive feedback on the manuscript.

L49: "changes in relative humidity is roughly steady" - that does not make sense. I think you mean that relative humidity reaches roughly a steady state value.

We agree your point. We revised the sentence as *"relative humidity reaches a steady state value"* in L46 in the revised version.

L90: "where the source and sink ... are identical" - I think you mean that they are in balance, not that they are identical.

In this sentence, we intend to describe identical sources of water vapour and heat. Sorry for the confusion caused. We have revised the sentence as *"the sources of water vapour and heat are identical"* in L81 in the revised version.

L92: Buoyancy is not driven by temperature gradients. It is driven by surface heating by absorption of solar radiation, which in turn results in temperature gradients and buoyancy.

Thank you for pointing out the erroneous description. We have revised the sentence as *"buoyancy driven by surface heating"* in L83 in the revised version.

L100: Please explain why you derive two equations for LE, and not just one. It is unclear what the benefit of having two equations is.

We agree with your concern. In order to clarify the benefit, we have revised the sentence as "*The two* equations (4) and (5) are complementary to each other in that they represent distinct thermodynamic paths, each of which will be discussed in the next section." in L111 in the revised version.

L190: "VPD budget" - VPD is not a conserved quantity, so there is no VPD budget.

Thank you for pointing out the erroneous term. We have revised the sentence as "Unlike many previous studies which focused on the steady state of VPD" in L176-177 in the revised version.

L230: You should be able to make a rough estimate from the canopy height about the significance of the heat storage term. This would support your argument.

Thank you for your suggestion. We added the following sentences in L217-219 in the revised version.

"For instance, Meyers and Hollinger (2004) showed that storage term comprised 14 % of net radiation for a maize field with a 3-m canopy height, and 8% of net radiation for a soybean field with a 0.9-m canopy height, implying larger heat storage capacities for taller crop canopies.